# Towards robust and generalizable representations of extracellular data using contrastive learning

**Ankit Vishnubhotla**[*]
Columbia University
New York
av3016@columbia.edu

**Charlotte Loh**[*]
MIT
Massachusetts
cloh@mit.edu

**Liam Paninski**
Columbia University
New York
liam@stat.columbia.edu

**Akash Srivastava**
MIT-IBM
Massachusetts
Akash.Srivastava@ibm.com

**Cole Hurwitz**
Columbia University
New York
ch3676@columbia.edu

## Abstract

Contrastive learning is quickly becoming an essential tool in neuroscience for extracting robust and meaningful representations of neural activity. Despite numerous applications to neuronal population data, there has been little exploration of how these methods can be adapted to key primary data analysis tasks such as spike sorting or cell-type classification. In this work, we propose a novel contrastive learning framework, **CEED** (**C**ontrastive **E**mbeddings for **E**xtracellular **D**ata), for high-density extracellular recordings. We demonstrate that through careful design of the network architecture and data augmentations, it is possible to generically extract representations that far outperform current specialized approaches. We validate our method across multiple high-density extracellular recordings. All code used to run CEED can be found at `https://github.com/ankitvishnu23/CEED`.

## 1    Introduction

High-density extracellular recordings now allow for simultaneous recordings of large populations of neurons across multiple brain regions with high temporal and spatial resolution. [1–4]. These large-scale recordings are essential for gaining insights into key biological processes such as vision, decision-making, and behavior which are distributed across brain regions [5]. Along with gaining insights into brain function, these technologies also promise to improve the scalability and accuracy of brain-computer interfaces which can restore motor function to paralyzed individuals [6].

A major bottleneck for interpreting neural population activity is processing the raw extracellular signal [7]. Although extracellular recordings contain a precise record of the coordinated neural activity, it must be extracted algorithmically through a processing step called *spike sorting*. A crucial assumption in spike sorting is that each recorded neuron has a unique spatiotemporal extracellular waveform based on its morphology and position relative to the recording device [8]. Using this unique identifier, it is possible to assign a detected extracellular action potential (spike) back to its putative neuron (unit). Along with spike sorting, another important task in extracellular analysis is cell-type classification. As neural circuits are diverse and heterogeneous, it is becoming increasingly important to profile detected units using their morphoelectrical features [9]. At a coarse level, it is

---

[*]Equal contribution

37th Conference on Neural Information Processing Systems (NeurIPS 2023).

possible to classify units as inhibitory or excitatory based on their extracellular profile [10]; however, it may be possible to divide units into finer subgroups [11–13].

For both spike sorting and cell-type classification, it is important to extract low-dimensional, meaningful features from extracellular waveforms. Despite the importance of feature extraction, current approaches are often ad hoc and lack robustness to common nuisance variables in extracellular recordings. By far the most common featurization method for spike sorting is principal components analysis (PCA) [3, 14–22]. Although PCA is relatively effective and scalable, it suffers from a few key drawbacks including: (1) a lack of robustness to extracellular nuisance variables such as spatiotemporally overlapping spikes (collisions), (2) an inability to model non-linear data, and (3) an objective function that aims to find features that explain variance rather than features that discriminate different waveforms. To improve the robustness of PCA, [21] introduced a supervised waveform denoiser that is able to 'clean' the waveforms before featurization. For morphoelectric cell-type classification, most feature extraction methods rely heavily on manually extracted features including action potential width, peak-to-peak amplitude, and the ratio of pre-hyperpolarization peak to the post-hyperpolarization peak [13]. Again, while scalable and effective, these features are too simple and ad hoc to fully capture morpho-eletrical differences [13]. Recently, a non-linear approach to cell-type classification, WaveMap, was introduced, utilizing UMAP [23] and Louvain community detection [24] to automatically find cell-type clusters [13].

In this work, we introduce a robust and generalizable feature learning method, CEED, for extracellular datasets. Our main hypothesis is that embeddings of extracellular waveforms that are invariant to both common and task-specific nuisance variables will be more useful for spike sorting and morphoelectric cell-type classification than current specialized feature extraction methods. For example, in extracellular recordings, there are a number of nuisance variables including collisions, correlated background noise, or variability in the time at which a spike is detected [21]. Each of these confounds pose a problem for traditional representation learning methods like PCA or manually extracted features. Along with these common nuisance variables, there are also task-specific nuisance variables such as the spatial position of detected spikes. For spike sorting, this information is crucial as spike locations are highly informative of neuron identity [25, 26]. However, for cell-type classification, neurons with different locations may still share a cell-type. In order to extract representations that are invariant to these nuisance variables, we utilize contrastive learning, which has been shown to approximately induce invariance in the representation space to a set of transformations [27]. We utilize a recent contrastive learning framework [28] for our training and implement a stochastic view generation module for extracellular waveforms. We validate our approach on multiple high-density extracellular recordings. Surprisingly, for cell-type classification, our representations appear to be more informative than state-of-the-art specialized methods even when performing zero-shot learning on an unseen animal and probe geometry. Our contributions are as follows:

1. We introduce a novel framework, CEED, for analyzing extracellular recordings based on invariance learning.

2. We implement a stochastic view generation module for both single-channel and multi-channel extracellular waveforms.

3. We demonstrate that CEED works well with multiple neural network architectures, including a novel transformer-based architecture with a spatiotemporal causal attention mask (SCAM).

4. We show that CEED outperforms specialized featurization methods for spike sorting and morphoelectric cell-type classification.

## 2  Background

**Contrastive Learning.** Contrastive representation learning [28] falls under a broad class of self-supervised learning methods [29–31] whose goal is to learn robust and generalizable representations by encouraging invariances to a prior-known set of transformations or nuisance variables. While contrastive learning has mainly been used to extract effective and transferable representations from image-based data [27, 28], more recently, it has also become a powerful tool in the sciences to learn invariances for physical systems [32, 33].

In computational neuroscience, representation learning has mostly been performed using generative models [34]. While this has led to many interesting insights into behavior [35] and decision-making [36], this paradigm is sensitive to nuisance variables and will not capture subtle changes in the

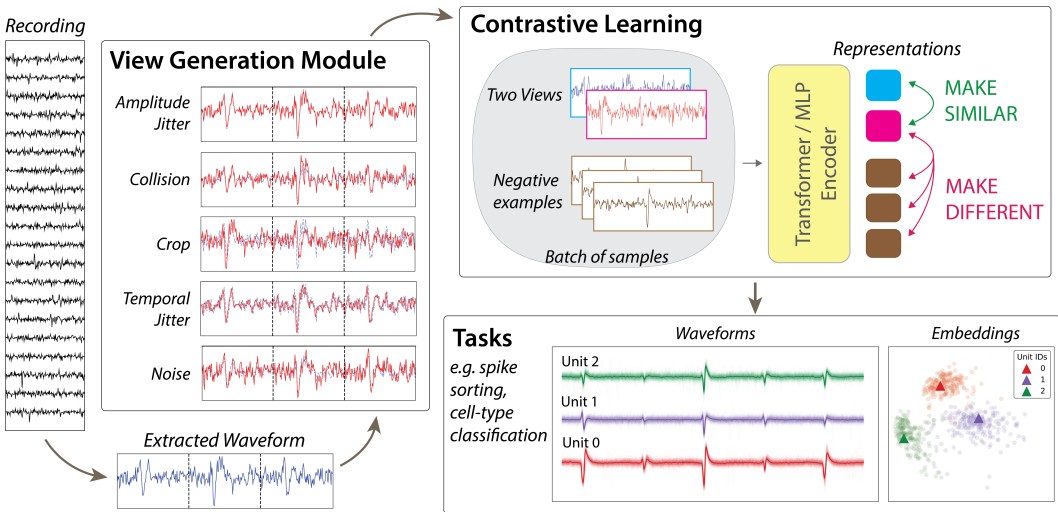

Figure 1: **CEED framework.** In CEED, we assume that waveforms are already extracted from an extracellular recording. Each waveform is then passed through our stochastic view generation module, where different views are obtained by applying transformations. These transformations induce a pre-defined set of invariances (see Section 3.2.1). Using these views, a neural-network based encoder, which can take the form of a multi-layer perceptron (MLP) or a transformer, is then trained to produce representations that respect the desired invariances. This is accomplished via contrastive learning, where representations from views of the same waveform are encouraged to be similar and views of different waveforms are encouraged to be dissimilar. When training completes, the learned representations can then be used for a series of downstream neuroscience tasks, such as spike sorting or morphoelectrical cell-type classification.

observation space that do not contribute much to the explained variance. While recent research into label-guided generative models [37–39] has somewhat alleviated these problems, it is still unclear if good generative performance is important for representation learning. Most recently, a number of contrastive learning methods have been introduced for learning robust and generalizable representations of neural population data [40–42]. To our knowledge, however, contrastive learning has not been applied to key primary data analysis tasks such as spike sorting or cell-type classification.

**Spike sorting.** In spike sorting, the goal is to extract a precise record of which neurons spiked at which time steps based on the raw extracellular data. For each electrode, the contributions of an unknown number of neurons are mixed together, making spike sorting a challenging blind-source separation task akin to the cocktail party problem [43]. While a number of spike sorting methods have been developed for microelectrode arrays (MEAs), almost all of these methods currently utilize PCA for feature extraction For example, the popular algorithms Klusta, HerdingSpikes2, Mountainsort4, SpykingCircus, Trideclous, and Kilosort all use a form of SVD/PCA in their processing pipelines. Despite the ubiquity of PCA as a feature extraction method for spike sorting, it lacks robustness to common nuisance variables in extracellular recordings, cannot capture non-linear features of the observed data, and prefers features that explain variance rather than those that discriminate different waveforms. More recently, it has been shown that non-linear autoencoders can have higher performance than PCA despite suffering from similar drawbacks [44].

**Cell-type classification.** Morphoelectric cell-type classification has many similarities to spike sorting. Similarly to spike sorting, you are also interested in grouping detected extracellular waveforms together. However, the granularity of these clusters should be lower, reflecting shared morphologies among the units in the recording. Classically, pre-defined morphoelectrical features (e.g., action potential width) were extracted from the waveforms to group together similar units [10, 45]. More recently, a non-linear method based on UMAP was introduced for cell-type classification [13]. In this work, it was shown that automatically discovering relevant features from extracellular waveforms can lead to more informative cell-type groupings than when simply clustering pre-defined features. It is important to note that, along with morphoelectric cell-type classification, it is also possible to classify cell-types by their function, morphology, physiology, molecular properties [46–48].

## 3 Contrastive Embeddings for Extracellular Data (CEED)

We introduce Contrastive Embeddings for Extracellular Data (CEED), a novel framework for extracting robust and generalizable representations of extracellular data via contrastive learning. CEED consists of three main components (see Figure 1) — (1) a stochastic view generation module that augments waveforms with both common and task-specific nuisance variables. (2) A neural network-based encoder that can extract low-dimensional representations from both single-channel and multi-channel waveforms. (3) A contrastive loss function that encourages the representations of views of the same waveform to be similar while forcing the representations of views of different waveforms to be dissimilar. We postulate that the representations extracted by CEED, which are approximately invariant to nuisance variables, can outperform specialized feature extraction methods for both spike sorting and cell-type classification.

### 3.1 Notation

To introduce our method, we must first define some notation for extracellular recordings. Let $P_M := \{p_m | m \in M\}$ be the position of all $M$ channels on the multi-electrode array (MEA), where each $p_m \in \mathbb{R}^3$ is the location of channel $m$. Now define $P_{\tilde{M}}$ to be the set of positions for a subset of channels, $\tilde{M} \subseteq M$, on the MEA.

Let $S := \{s_n\}_{n=1}^N$ be the set of $N$ spiking events that are detected in an extracellular recording. Now, let the extracellular waveform of a spiking event $s_n$ on a channel $m$ be defined as $W_{n,m} := \{r_{n,m}^{(0)}, r_{n,m}^{(1)}, ..., r_{n,m}^{(T)}\} \in \mathbb{R}^T$, where $T$ contiguous samples from the recording define a waveform and $r_{n,m}^{(t)}$ is the value of the waveform at sample $t$. This can be thought of as a window of samples of size $T$ on channel $m$ which includes the spike $s_n$.

The amplitude of a spike $s_n$ on a channel $m$ can be defined as $A_{n,m} := \max_t |W_{n,m}|$. This is also known as the absolute maximum voltage recorded on channel $m$. The timestep $t$ at which the absolute maximum voltage occurs can vary for each channel. Therefore, for a spike, we center the waveform on each channel using the timestep at which the maximum $A_{n,m}$ occurred (i.e., the timestep of the maximum amplitude channel). For all spikes in the recording, waveforms are aligned in this manner so that the amplitude on the maximum amplitude channel occurs at the same timestep $t$.

Let $\mathbf{W}_{n,\tilde{M}} \in \mathbb{R}^{T \times |\tilde{M}|}$ be the set waveforms for a spike on a subset of channels and let $\mathbf{A}_{n,\tilde{M}} \in \mathbb{R}_{>0}^{|\tilde{M}|}$ be the set amplitudes for a spike on a subset of channels, where $|\tilde{M}|$ is the number of channels.

### 3.2 Stochastic view generation module

#### 3.2.1 Common invariances

Let us define the representation of the waveform $\mathbf{W}_{n,\tilde{M}}$ for spike $s_n$ as $\mathbf{z}_n \in \mathbb{R}^D$. There are a number of nuisance variables that change the observed waveform of a spike without changing the underlying signal. Therefore, we can define a set of invariances we wish to impose on our representations,

1. **Amplitude voltage jitter** - We want representations that are invariant to "small" amplitude variability. To this end, if we scale the amplitude of the waveform $\mathbf{W}_{n,\tilde{M}}$ for spike $s_n$ such that $\widetilde{\mathbf{W}}_{n,\tilde{M}} = \mathbf{W}_{n,\tilde{M}} \times (1 \pm \epsilon)$, then the representation $\tilde{\mathbf{z}}_n$ for $\widetilde{\mathbf{W}}_{n,\tilde{M}}$ should be such that $\tilde{\mathbf{z}}_n = \mathbf{z}_n$.
2. **Correlated background noise** - We want representations that are invariant to the correlated background noise found in real extracellular recordings. We assume extracellular noise can be modeled as a spatiotemporal matrix Gaussian $\mathcal{MN}(0, U, V)$, where $U \in \mathbb{R}^{T \times T}$ models the temporal correlation of a waveform and $V \in \mathbb{R}^{|M| \times |M|}$ models the spatial correlation between channels. For a spike $s_n$, we want a representation such that if we sample $n \sim \mathcal{MN}(0, U, V)$ and form $\widetilde{\mathbf{W}}_{n,\tilde{M}} = \mathbf{W}_{n,\tilde{M}} + n_{\tilde{M}}$, where $n_{\tilde{M}}$ is the noise on the channels subset $\tilde{M}$, then $\widetilde{\mathbf{W}}_{n,\tilde{M}}$ should still yield a representation $\tilde{\mathbf{z}}_n = \mathbf{z}_n$.
3. **Spike collisions** - We want representations that are invariant to the voltage distortions caused when spikes "collide" spatiotemporally. A collision is when two or more spikes occur at similar times in the same location on the probe leading to observed waveforms that are distorted. To this end, consider two spikes $s_{n_1}$ and $s_{n_2}$ with waveforms $\mathbf{W}_{n_1,\tilde{M}_1}$ and $\mathbf{W}_{n_2,\tilde{M}_2}$ such that $\tilde{M}_1$

and $\tilde{M}_2$ share some number of channels. Now let us shift $s_{n_2}$ by $k$ time steps ($-T/2 < k < 0$ or $0 < k < T/2$) such that for any channel $m \in \tilde{M}_2$, $W_{n2,m} := \{r_{n_2,m}^{(0+k)}, r_{n_2,m}^{(1+k)}, ..., r_{n_2,m}^{(T+k)}\}$. We refer to this temporally shifted waveform as $\mathbf{W}_{n_2,\tilde{M}_2}^{\text{shift}}$. A new waveform that is a linear sum of these two waveforms $\widetilde{\mathbf{W}}_{n,\tilde{M}_1} = \mathbf{W}_{n_1,\tilde{M}_1} + \mathbf{W}_{n_2,\tilde{M}_2}^{\text{shift}}$ should yield a representation $\tilde{\mathbf{z}}_n = \mathbf{z}_{n_1}$.

4. **Channel subsets** - We want representations that are invariant to changes in the subset of channels $\tilde{M}$ used to define $\mathbf{W}_{n,\tilde{M}}$. So for a new subset of channels $\tilde{M}'$, as long as the channel with the highest amplitude is still contained in $\tilde{M}'$, then the waveforms $\mathbf{W}_{n,\tilde{M}'}$ should still yield a representation $\tilde{\mathbf{z}}_n = \mathbf{z}_n$.

### 3.2.2 Task-specific invariances

While all feature extraction methods suffer from common extracellular nuisance variables, there are also task-specific nuisance variables. For example, the position and orientation of neurons in the recording are essential information for spike sorting. However, for cell-type classification, this information can be a confound when trying to find shared morphoelectrical features. Therefore, we propose an additional set of invariances for cell-type classification,

1. **Cell position** - For cell-type classification, we want representations that are invariant to the channel positions $P_M$ at which a spike is detected. So for a waveform $\mathbf{W}_{n,M}$, any uniform changes to the channel positions $P_{\tilde{M}}$ (without changing the waveforms at each channel) should still yield a representation $\tilde{\mathbf{z}}_n = \mathbf{z}_n$.

2. **Cell amplitude** - As amplitude is mainly a function of cell position and orientation, we want representations that are invariant to any uniform changes in the amplitudes. To this end, for any positive value $a$, a change in the amplitudes for the spike $s_n$, $\widetilde{\mathbf{W}}_{n,\tilde{M}} = \mathbf{W}_{n,\tilde{M}} \times a$, should still yield a representation $\tilde{\mathbf{z}}_n = \mathbf{z}_n$.

To achieve these task-specific invariances for cell-type classification, we directly transform the training data. First, we extract the max channel waveform for each spike. Then, we normalize each waveform to be between -1 and 1, thus removing any positional information [13]. A drawback of this approach is that we discard multi-channel information that may be useful for classifying different cell-types [12] (see Section 7 for a more detailed discussion). The full view generation pipeline is detailed in Supplementary Materials Section A.1.

## 3.3 Encoder Architecture

For the encoder of CEED, we explored two neural network architectures – a transformer-based network with a novel Spatiotemporal Causal Attention Mask (SCAM) and a simple multi-layer perceptron (MLP) which is a more computationally efficient alternative (i.e., can fit on a single GPU). For a runtime comparison of these two architectures, see Supplementary Materials Section E.

### 3.3.1 Transformer-based encoder with spatiotemporal causal attention mask (SCAM)

Our first proposed architecture utilizes transformers [49] which have been highly successful across a series of tasks in natural language processing [50, 51]. Transformers have a natural inductive bias towards time-series and sequence-based data and are thus highly suitable for extracellular waveforms. For this work, we designed a novel spatiotemporal causal attention mask (SCAM) to obey causality across time and channels. Specifically, we allow every recorded time step in a waveform to attend to time steps on other channels as long as those data points precede it in time. Full details and visualization of the transformer-based architecture, including implementation details are available in Supplementary Materials Section A.2.

### 3.3.2 Multi-layer perceptron (MLP)

While a transformer-based architecture provides a natural inductive bias towards time-series data appropriate for extracellular recordings, a downside to such an architecture is its high computational complexity and the requirement that it needs multiple GPUs to train. To demonstrate the generality of the CEED framework, we also propose a simpler MLP architecture that can be trained on a single GPU. The MLP encoder is a straightforward model that consists of three layers with sizes $[768, 512, 256]$ and ReLU activations between them.

## 3.4 Objective function

The encoder is trained as follows. Let $\tilde{\mathbf{z}}_i$ and $\mathbf{z}_i$ be representations of the two views of the input waveform $i$, where the two views differ by the invariances listed in Sections 3.2.1 and 3.2.2. The optimization objective for a batch of $B$ samples follows the contrastive loss utilized by SimCLR [28] and several other representation learning methods [52, 53];

$$\mathcal{L} = \sum_{i=1}^{B} -\log \frac{\exp(\hat{\mathbf{z}}_i \cdot \hat{\tilde{\mathbf{z}}}_i)/\tau}{\sum_{k \neq i} \exp(\hat{\mathbf{z}}_i \cdot \hat{\mathbf{z}}_k/\tau)} \tag{1}$$

where $\hat{\mathbf{z}}_i$ and $\hat{\tilde{\mathbf{z}}}_i$ are the L2 normalized representations and $\tau$ is a temperature hyperparameter. Following [28], we also include a 2-layer MLP projector network after the encoder and the loss function operates on the outputs of this projector network (see Supplementary Materials Section A.3 for details and ablations of the projector network and its architecture).

# 4 Datasets

To train and evaluate our model, we make use of two publicly available extracellular recordings published by the International Brain Laboratory (IBL): the DY016 and DY009 recordings [54]. These multi-region, Neuropixels 1.0 recordings are taken from a mouse performing a decision-making task (see Supplementary Materials Section C for more details).

## 4.1 Spike sorting

To evaluate how useful the features learned by CEED are for spike sorting, we constructed three datasets using units found by Kilosort 2.5, a full spike sorting pipeline manually tuned by IBL. The first dataset was extracted from the DY016 extracellular recording. It consisted of a 10 unit train and test dataset where all 10 units were classified as "good" by IBL's quality metrics [55]. We selected these units for their high waveform diversity (qualitatively) and because they had a relatively high amplitude, i.e., peak-to-peak (ptp). For this dataset, we constructed training sets of 200 or 1200 spikes per unit with a test set of 200 spikes per unit. For each spike, we extracted waveforms from 21 channels centered on the maximum amplitude channel. Although we extract 21 channels for our data augmentations (see Supplementary Materials Section A.1), we train and evaluate our model (and baselines) on either 5 or 11 channel subsets.

The second dataset was extracted from both the DY016 and DY009 extracellular recordings. This dataset consisted of a 400 training units and 10 test units (the same 10 units evaluated in the first dataset). We extracted 200 units from each recording to create the training set with 200 or 1200 training spikes per unit. The goal of this dataset was to test how well CEED could generalize across a large set of units with varying quality. To this end, we also evaluated the performance of CEED on 100 random test sets of varying sizes (see Supplementary Materials Section D).

The third and final dataset was also extracted from both the DY016 and DY009 extracellular recordings. Unlike the first and second dataset, where the training set contained spikes from the test units (in-distribution), we purposefully excluded all units in the test set from the training set. To this end, this dataset consisted of 390 training units and 10 test units (the same 10 units evaluated in the above datasets). For all units in the training set, we utilized either 200 or 1200 training spikes per unit. The goal of this dataset was to test how well CEED could generalize to out-of-distribution (OOD) units.

## 4.2 Cell-type

For our cell-type classification dataset, we utilized the DY016 extracellular recording. We extracted the same 10 IBL "good" units used for the spike sorting datasets. To remove positional information from each spike, we only extract waveforms from the maximum amplitude channel and we normalize each waveform as described in Section 3.2.2.

# 5 Experiments

## 5.1 Spike sorting

**Re-sorting units extracted using KS2.5** In this experiment, we aim to demonstrate that CEED extracts more useful features for spike sorting than both PCA and a non-linear autoencoder. We evaluate these methods on both an in-distribution (ID) and out-of-distribution (OOD) waveform discrimination task. Specifically, we train CEED, PCA, and an autoencoder on all three spike sorting datasets (see Section 5) and perform inference on the spikes from the test units. Then, we 're-sort' these test spikes back to their putative units by performing clustering on the resulting embeddings. For clustering, we use a parametric clustering algorithm, the Gaussian Mixture Model (GMM), and a non-parametric clustering algorithm, HDBSCAN [56]. To evaluate how well each clustering corresponds to the ground-truth, we use the Adjusted Rand Index (ARI) [57]. One can also compute the accuracy after optimal permutation (e.g., Hungarian algorithm), however, this is will give similar results to the ARI. To strengthen our spike sorting baselines, we also compare CEED to PCA and an autoencoder trained and tested on denoised waveforms using the YASS waveform denoiser [21, 58]. The YASS denoiser was trained to denoise single channel waveforms so we apply it independently to each channel and it improves clustering of PCA across the board. For all baselines, we sweep across (3,5,7,9) principal components and 3-11 channel subset sizes.

## 5.2 Cell-type classification

**Cell-type classification of IBL recordings** In this analysis, we aim to demonstrate that CEED can extract putative cell-type clusters from IBL extracellular recordings. For the IBL recordings, we extract average waveforms (templates) for 163 good units pooled over DY016 and DY009. We then run inference on the extracted templates using CEED and perform a GMM clustering on the resulting embeddings to find putative cell-types. As there is no ground-truth for the IBL recordings, we validate our results by visualizing the templates for each cell-type cluster.

**Zero-shot cell-type classification.** In this experiment, we compare the inferred representations of CEED to a state-of-the-art cell-type classification method, WaveMap, on an out-of-distribution (OOD) dataset. Not only is this dataset not seen during training, but both the animal and the probe geometry are completely different than the dataset used to train CEED. The dataset consists of 625 templates (average waveforms) extracted from units which are recorded while a monkey performs a discrimination task [13]. The probe used to record these waveforms is a Plexon U-probe. The goal of this experiment is to find putative cell-types that explain the waveform variability seen in the data. We propose to run CEED on the 625 templates and then perform a GMM clustering of the resulting embeddings. We choose the number of clusters by again utilizing the Elbow method and BIC. Given how OOD this dataset is compared to the recordings used to train CEED, success on this task would demonstrate the robustness and generalizability of CEED.

# 6 Results

## 6.1 Spike Sorting

**Re-sorting units extracted using KS2.5** For our re-sorting task, we find that CEED outperforms both PCA and the non-linear autoencoder using raw waveforms or denoised waveforms across all three datasets introduced in Section 4. The results for this analysis can be found in Table 1. As can be seen, in both the ID and OOD regimes, CEED has much higher performance than both baseline models. For all methods we utilize the same number of latent dimensions for the analysis (5D). We also show in Supplementary Materials Section D that the strong performance of CEED generalizes to many different sets of test units.

Along with this results table, we also quantitatively and qualitatively compare the performance of CEED to PCA on the 10 neuron train and test dataset in Figure 2. Visually, it can be seen that CEED's features are far more informative about each unit's identity than the representations found by PCA. The performance of CEED is also much higher when being clustered by either HDBSCAN or a GMM even when when more principal components are afforded for the analysis.

| Method | 10 neuron train set 10 neuron ID (ARI) | 400 neuron train set 10 neuron ID (ARI) | 390 neuron train set 10 neuron OOD (ARI) |
|---|---|---|---|
| CEED (1200 spikes, 11 channels) | **.89** ± **.04** | **.79** ± **.09** | **.78** ± **.05** |
| CEED (1200 spikes, 5 channels) | .83 ± .03 | .76 ± .07 | .77 ± .08 |
| Denoised PCA (1200 spikes, 11 channels) | .39 ± .05 | .45 ± .04 | .46 ± .04 |
| Denoised PCA (1200 spikes, 5 channels) | .46 ± .07 | .48 ± .04 | .49 ± .04 |
| Autoencoder (1200 spikes, 11 channels) | .47 ± .06 | .35 ± .03 | .28 ± .02 |
| Autoencoder (1200 spikes, 5 channels) | .43 ± .06 | .37 ± .03 | .33 ± .01 |

Table 1: **Benchmarking CEED, PCA, and an autoencoder on in-distribution (ID) and out-of-distribution (OOD) data.** For evaluation, we fit 50 GMMs to the embeddings and compute the mean and std. of the adjusted rand index (ARI). First column: we train and test each method with spikes from 10 neurons. Second column: we train each method with spikes from 400 neurons and then test on the original 10 neurons which are **included** in the training set (ID). Third column: we train each method on spikes from 390 neurons and test on the original 10 neurons which are **not included** in the training set (OOD). This experiment demonstrates that CEED peforms well on OOD data and can outperform a non-linear autoencoder. All CEED results are generated using the MLP encoder.

## 6.2 Cell-type Classification

**Cell-type classification of IBL recordings** The results for cell-type classification of the pooled IBL recordings are shown in Supplementary Materials Section B. We choose the number of cell-type clusters by sweeping over 1-10 clusters and choosing the minimum BIC. We find that 4 clusters explain most of the waveform variability. Interestingly, we find good separation in CEED's embedding space between narrow-spiking and broad-spiking units which indicates that we may be able to discriminate between inhibitory and excitatory subtypes.

**Zero-shot cell-type classification.** The results for cell-type classification of OOD single unit data are shown in Figure 3. Despite training on IBL extracellular datasets from a mouse brain recorded with Neuropixels 1.0, CEED is able to generalize to unseen data from a completely different animal (monkey) and probe (Plexon U-probe). In Figure 3A, we visualize the inferred representations from CEED using a 2D UMAP and by coloring the points with the output of a GMM that is trained on the 5D contrastive representations. We choose the number of cell-type clusters by sweeping over 1-10 clusters and then using the Elbow Method on the BIC curve. With this method, we discover 6 putative cell-types (similarly to WaveMap). Upon visual inspection, our cell-types are more well-isolated from each other than those of WaveMap (Figure 3C). To quantitatively assess which cell-type classification method better reflects the 'real' differences in extracellular waveforms, we utilize a supervised classifier which is trained to predict cell-type labels using input waveforms (introduced in [13]). In Figure 3B and Figure 3D, we show the results of this data on 5 cross-validation folds. Despite never training on this dataset, CEED has a much higher accuracy (93.4%) compared to that of WaveMap (88.8%). A small caveat is that we ran WaveMap using publically available code but were unable to precisely reproduce the original result in the paper which finds 8 cell-type clusters. Despite this difference, the accuracy value reported in the WaveMap paper for the 8 cell-type clusters (91%) was still lower than that of CEED.

## 7 Discussion

In this paper, we introduced a novel representation learning method, CEED, for extracellular recordings. Our main hypothesis was that by finding representations of extracellular waveforms that are robust to both common and task-specfic nuisance variables, we can outperform specialized feature extraction approaches on two key tasks: spike sorting and morphoelectrical cell-type classification. We validate CEED on multiple high-density extracellular datasets. For spike sorting, we show that CEED extracts features that far outperforms those of PCA and a non-linear autoencoder on a waveform discrimination task. For cell-type classification, we show that CEED is able to extract discriminative features of extracellular waveforms that allow for finding morphological subgroups in an unsupervised manner. Surprisingly, we find that CEED even outperforms a recent non-linear cell-type classification method WaveMap, on an animal and probe geometry unseen during training.

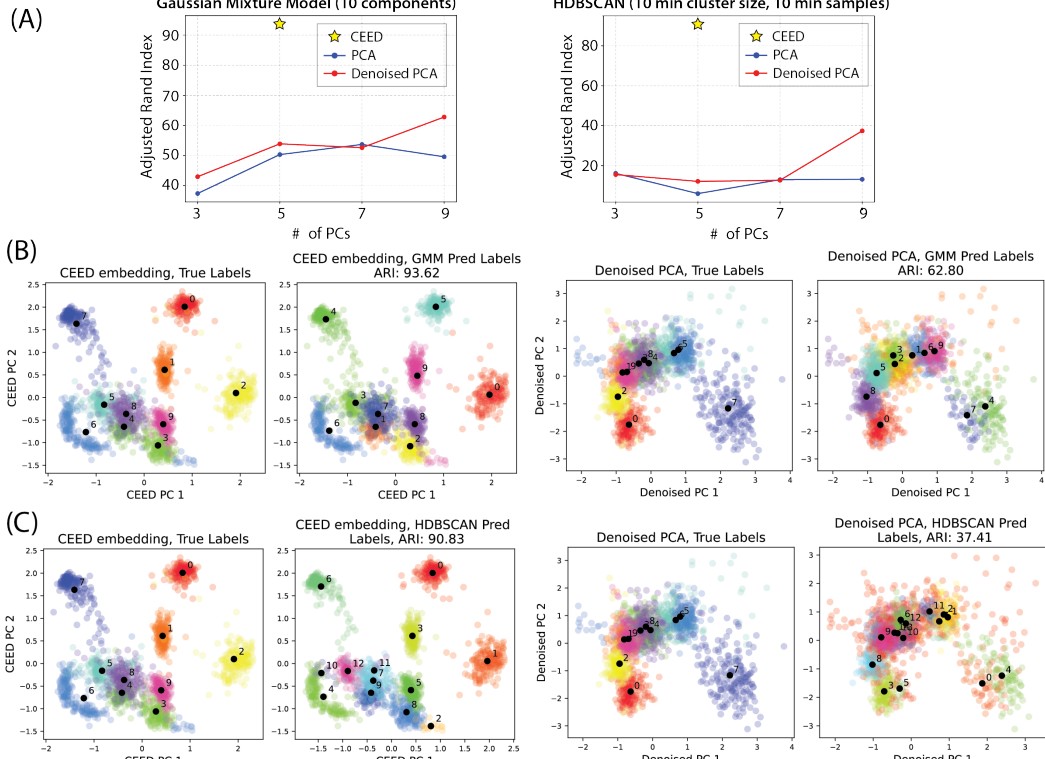

Figure 2: **CEED outperforms PCA on spike sorting featurization**. Here, we show results for CEED (using the SCAM architecture) and PCA when trained and evaluated on a 10 neuron dataset. (A) Clustering results for the two featurization methods using a parametric (GMM) and non-parametric (HDBSCAN) clustering algorithm. As can be seen, the featurization learned by CEED is much more discriminative than that of PCA for both clustering methods even when using a supervised denoiser to reduce noise in the data. (B) Visualized results of a 10 component GMM clustering on the learned embeddings from CEED (left) and denoised PCA (right). (C) Visualized results of the HDBSCAN clustering applied to the learned embeddings from CEED (left) and denoised PCA (right).

Despite the significant performance improvements of CEED, there are few limitations that must be addressed before it can become a plug-and-play method for extracellular analysis. Firstly, the training sets we utilize in this work are still quite small and lack neuron diversity. For CEED to be generalizable to multiple recordings and animals, more diverse datasets must be constructed. Secondly, the best results of CEED are obtained when using the transformer-based encoder which requires multiple GPUs and is currently quite slow to train. Recent progress in acceleration software [59] offer promising solutions to speed up computation and incorporating these methods into CEED could be a future direction. Thirdly, all spike sorting results in the paper are from re-sorting already sorted datasets; CEED must be incorporated into a full spike sorting pipeline in order to be used by many different research groups. Finally, our cell-type results do not include any functional classification which could help validate the clusters found by CEED.

## 8 Broader Impact

Although CEED has the potential to improve key tasks in extracellular analysis, a drawback of our approach compared to simple approaches like PCA is the additional computational resources it requires to train and run. Contrastive learning requires large batch sizes to achieve state-of-the-art performance which means, if training the SCAM transformer model, we often have to run experiments on large-scale, multi-gpu clusters. Moreover, transformer-based architectures have hundreds of thousands of parameters, which further increases CEED's compute requirements. As highly parameterized deep neural networks produce large amounts of carbon emissions [60], CEED could have a possible negative impact on the environment. We also propose an MLP-based

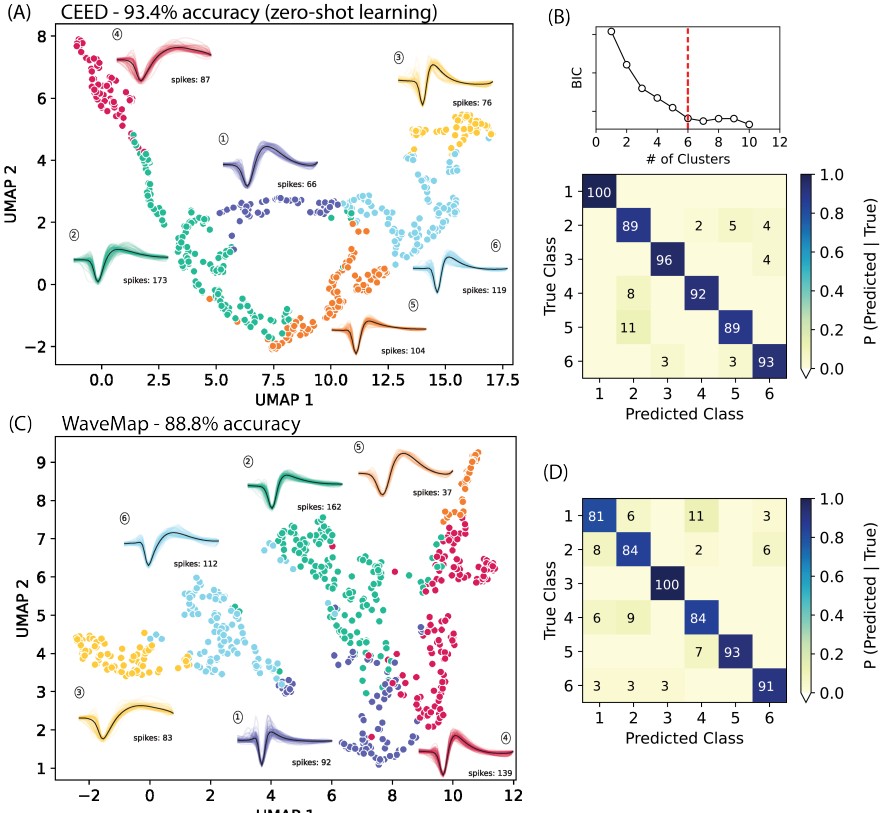

Figure 3: **CEED outperforms WaveMap on morphoelectrical cell-type classification with zero-shot learning.** (A) A 2D UMAP embedding of CEED's inferred representations for a OOD dataset which consists of 625 single units extracted from a monkey recording. We cluster the representations using a GMM and plot each cluster with a distinct color. For each cluster, we also plot the waveforms associated with the discovered cell-type. It can be seen that narrow-spiking waveforms (3,5,6) are well-separated from broad-spiking waveforms (1,2,4) which may indicate good separation of inhibitory and excitatory subtypes, respectively. Surprisingly, CEED is trained on a different animal and probe geometry, but can still generalize to this dataset, outperforming WaveMap on a classification baseline introduced in [13]. (B) On the top, we demonstrate how we chose the number of clusters for the GMM, i.e., with the Elbow Method and BIC. On the bottom, we show the confusion matrix of a gradient boosted decision tree classifier trained to map raw waveforms to the cell-type clusters extracted by CEED. The accuracy of each method is defined as the average of the diagonal. (C) A 2D UMAP embedding the single unit dataset using WaveMap. The clusters are colored according to WaveMap's outputted labels. Narrow-spiking clusters (1,2,4) are more mixed with broad-spiking clusters (3,5,6). (D) The confusion matrix of a gradient boosted decision tree classifier trained to map raw waveforms to the cell-type clusters extracted by WaveMap.

architecture that performs comparably to a transformer while only using a single GPU, which can lower CEED's impact on the environment.

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
