# Supplementary Materials: Towards robust and generalizable representations of extracellular data using contrastive learning

**Ankit Vishnubhotla**[*]
Columbia University
New York
av3016@columbia.edu

**Charlotte Loh**[*]
MIT
Massachusetts
cloh@mit.edu

**Liam Paninski**
Columbia University
New York
liam@stat.columbia.edu

**Akash Srivastava**
MIT-IBM
Massachusetts
Akash.Srivastava@ibm.com

**Cole Hurwitz**
Columbia University
New York
ch3676@columbia.edu

## A  Implementation details of CEED

### A.1  Stochastic data augmentation module in CEED

The CEED stochastic data augmentation pipeline consists of the following transformations that are applied to the waveforms during training of the contrastive model,

1. **Amplitude Jitter**: For the amplitude jitter augmentation we randomly sample a value from a uniform random variable $x \sim Uniform(0.9, 1.1)$ and multiply the entire waveform by $x$. This augmentation is applied to waveforms with a probability of 0.7.

2. **Temporal Jitter**: This augmentation works through two steps. In the first step we upsample the original waveform by 8 times on every channel and then select the same interpolation of the waveform on every channel. More specifically, for an upsampled waveform on a channel we select a starting sample for the waveform in the range 1 to 8 and then select every sample offset by 8 from it so that the number of samples for the waveform is the same as the original waveform. We then replicate this on every channel (if the waveforms are multi-channel) with the same starting offset for each channel. Then we shift the new waveform two samples forwards or backwards to provide a temporal shift as well. The first part of the temporal jitter is adapted from the YASS pipeline [3]. This augmentation is applied to waveforms with a probability of 0.6.

3. **Noise**: For the noise augmentation we describe the noise as being sampled from a spatiotemporal matrix Gaussian $\mathcal{MN}(0, U, V)$, where $U$ models the temporal covariance and $V$ models the spatial covariance. We then select the appropriate channels from the sampled noise such that the noise has the same shape as the waveform and add this to the waveform. We model the spatiotemporal aspects of the noise by finding noise segments in the channel recordings and computing the spatial and temporal variances of the noise, which is adapted from the YASS pipeline [3]. This augmentation is applied to waveforms with a probability of 0.5.

4. **Collisions**: This augmentation randomly selects a waveform from the training set, scales it uniformly by a value $x \sim Uniform(0.2, 1)$, randomly shifts it forwards or backwards by a number of samples dictated by $s \sim Uniform(5, 60)$, and finally adds this random scaled, shifted waveform to the original waveform. The values 5 and 60 are calculated by rounding

37th Conference on Neural Information Processing Systems (NeurIPS 2023).

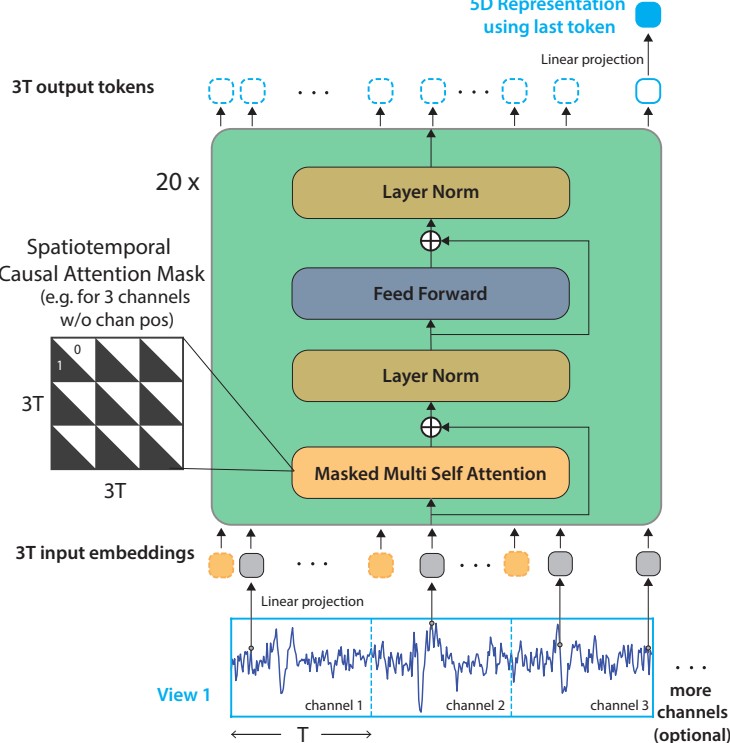

**Supplementary Figure 1: SCAM.** A transformer-based encoder is used to obtain representations for each waveform. Each time point on the waveform is projected into a 64-dimensional embedding and fed into a transformer-based architecture and the representation is obtained using a linear projection from the last output token of the transformer module. CEED uses a spatio-temporal causal attention mask (SCAM) to obey causality; points can attend across channels, but only to points that were recorded before them.

4% and 50% of the total number of samples in the waveform. This augmentation is applied to waveforms with a probability of 0.4.

5. **Crop**: The crop augmentation takes in a large number of channels and then subselects a smaller contiguous window of channels from the original set. When subsampling channels, the probability that the subset is centered on the max amplitude channel is 0.5. The probability that the subset is centered on a channel that is not the max channel is also 0.5, however, the new central channel must still allow for extracting a subset of channels that contains the original max amplitude channel. In this case, the channel that is selected as the new center channel is chosen uniformly at random from all eligible channels.

## A.2 Transformer-based encoder with spatiotemporal causal attention mask (SCAM)

For the transformer-based encoder, we used an architecture adapted from the language model GPT-2 [4]. Specifically, we modified the original model and instead use 20 layers, 4 attention heads, and a hidden size of either 64 (for multi-channel data) or 32 (for single channel data). Overall, our network has approximately 1M parameters (for multi-channel data) and 0.5M (for single channel data). Our model operates on the data by linearly projecting each data point to an embedding of dimension 64 (32 for single channel models) and then feeding the embedding into into the encoder which contains the transformer blocks. The final representation is the linear projection of the last output token onto a 5-dimensional representation space (for both the multi-channel and single-channel datasets). Details of this architecture is shown in Supplementary Figure 1.

**Multi-channel waveform models.** Following [2], we include a 2-layer MLP projector network after the transformer encoder with hidden dimensions 512 and 128 and a projection dimension of 5. The loss function, Equation (1) in the main text, is computed on the projection embeddings and representations are extracted from before the projector network similar to the default set up in [2]. We use a batch size of 128 and learning rate of 0.0001 for all multi-channel models. This learning rate is set by optimizing via grid search on a held out validation set on the DY016 10-neuron dataset and the same learning rate is used for all models and tasks.

**Single-channel waveform models.** We also include a 2-layer MLP network after the transformer encoder with similar hidden dimensions as above. Instead of extracting representations from before the MLP, we found that using the 5-dimensional MLP output resulted in slightly better performances than using the representations before the MLP. We thus use the output of the MLP as the representations in the single-channel waveform models. The input to the MLP (after linear projection from the last output token) is set to be 128-dimensional. For single channel models, we use a batch size of 512 and a learning rate of 0.001. Again this learning rate was determined via a grid search on a held-out validation set on the DY016 10-neuron dataset and subsequently fixed for all experiments.

### A.3 Projector network details and ablations

The SimCLR training scheme utilizes a projector network that takes in the output of the encoder network as input and its output functions as the representation on which the loss operates [2]. The CEED projection network includes 2 layers that are 512 nodes each. For results in the paper that use the MLP we use the representations of waveforms after the projector for all analysis. Ablating the projector network for the final representation does impact performance, as can be seen in Supplementary Table 1. We can see that adding the projector layers to the encoding of the waveforms helps the clustering of such waveforms by a noticeable amount.

| 400 neuron, 200 spikes train set | CEED (11 channels) no projector | CEED (11 channels) projector | CEED (5 channels) no projector | CEED (5 channels) projector |
|---|---|---|---|---|
| GMM | $.75 \pm .06$ | $\mathbf{.82 \pm .09}$ | $.65 \pm .05$ | $.73 \pm .08$ |
| KNN | .93 | **.96** | .90 | **.96** |

Supplementary Table 1: **Projector ablations**. For each column, we benchmark an MLP CEED model with or without the projector network used during training on the same 10 neurons as the paper Table 1, Column 2. We train CEED with 200 spikes from the same 400 neurons used in previous tables for this analysis. Here we can see that the projector improves performance for both the 5 channel and 11 channel MLP CEED models.

### A.4 Hyperparameter Search

We performed a search over the various hyperparameters used in CEED including MLP architecture, representation dimension, learning rate, and batch size. For the MLP architecture search we trained and tested MLP architectures with 2, 3, and 4 fully connected layers. In Supplementary Table 2 we can see that the different architectures do not influence the final results greatly, and that the 3-layer architecture used in the paper performs comparatively well. The results for a small learning rate search also show a similar story, with all three learning rates yielding very similar performance (Supplementary Table 3). In terms of batch size we also see that there is not much effect when modulating this hyperparameter (Supplementary Table 4), and that the batch size of 512 used in the CEED benchmark models seems appropriate. We also modulate the representation dimension of the CEED models and see a larger impact when CEED is trained to produce more compact (low d) representations, but that performance saturates at a representation dimension of 5 in both of our metrics (Supplementary Table 5). Overall, the hyperparameters chosen for the CEED models that we use for benchmarking all seem appropriate and produce more optimal performance. Moreover, we see that the CEED model and training scheme itself is robust and not overly sensitive to any of the hyperparameter searches we performed.

| CEED (11 channels, 200 spikes) | 2 | 3 (used) | 4 |
|---|---|---|---|
| GMM | $.83 \pm .09$ | $.82 \pm .07$ | $.82 \pm .09$ |
| KNN | .96 | .96 | .96 |

Supplementary Table 2: **MLP Architecture Search**. For each column, we benchmark an MLP CEED model with the specific number of layers denoted on the same 10 neurons as paper Table 1, Column 2. We train CEED with 200 spikes from the same 400 neurons used in previous tables for this analysis. The models used for results in the paper use a 3 layer architecture. Here we can see that the range of hidden layers for the MLP architecture used do not change the results significantly.

| CEED (11 channels, 200 spikes) | 1e-4 | 1e-3 (used) | 5e-3 |
|---|---|---|---|
| GMM | $.81 \pm .10$ | $.82 \pm .07$ | $.81 \pm .06$ |
| KNN | .97 | .96 | .95 |

Supplementary Table 3: **Learning Rate Search**. For each column, we benchmark an MLP CEED model with the specific learning rate on the same 10 neurons as paper Table 1, Column 2. We train CEED with 200 spikes from the same 400 neurons used in previous tables for this analysis. The models used for results in the paper use a learning rate of 1e-3. Here we can see that the range of learning rates used do not shift the results significantly. These CEED results are generated using an MLP architecture.

| CEED (11 channels, 200 spikes) | 128 | 256 | 512 (used) |
|---|---|---|---|
| GMM | $.80 \pm .09$ | $.79 \pm .08$ | $.82 \pm .07$ |
| KNN | .95 | .96 | .96 |

Supplementary Table 4: **Batch Size Search**. For each column, we benchmark an MLP CEED model with the specific training batch size denoted on the same 10 neurons as in the main text Table 1, Column 2. We train CEED with 200 spikes from the same 400 neurons used in previous tables for this analysis. The models used for results in the paper use a batch size of 512 during training. Here we can see that the batch sizes do not change the results signficantly, but that the higher batch size seems to benefit CEED.

| CEED (11 channels, 200 spikes) | 2D | 3D | 4D | 5D (used) | 6D | 10D |
|---|---|---|---|---|---|---|
| GMM | $.44 \pm .01$ | $.67 \pm .02$ | $.80 \pm .06$ | $.82 \pm .07$ | $.79 \pm .09$ | $.68 \pm .04$ |
| KNN | .62 | .88 | .94 | .96 | .96 | .96 |

Supplementary Table 5: **Representation Dimension Search**. For each column, we benchmark an MLP CEED model with the specific representation dimension (K) on the same 10 neurons as paper Table 1, Column 2. We train CEED with 200 spikes from the same 400 neurons used in previous tables for this analysis. The models used for results in the paper use a representation dimension of 5. Here we can see that a representation dimension of 5 yields the best results on both the GMM and KNN metrics. These CEED results are generated using an MLP architecture.

### A.5 Runtimes of SCAM vs MLP

Supplementary Table 6 compares the run-times between the transformer-based encoder (SCAM) and the simple MLP encoder for a variety of settings with different neuron and channel number. SCAM uses 16 GPUs in parallel while the MLP runs on a single GPU. Yet, the runtimes of the MLP is significantly lower, thus showing that the MLP encoder provides an effective alternative if computational resources is a constraint.

| Architecture | 10n, 5chan, 1200s | 10n, 11chan, 1200s | 400n, 11chan, 200s | 400n, 11chan, 1200s |
|---|---|---|---|---|
| SCAM | 35s | 96s | 186s | — |
| MLP | 10s | 10s | 29s | 185s |

Supplementary Table 6: **Per epoch runtime of CEED**. SCAM: the reported times are with 16 NVIDIA V100s in parallel. MLP: the reported times are with 1 NVIDIA V100.

# B Cell-type classification

## B.1 Pooled IBL recordings

In this experiment, we perform cell-type classification on 163 good units extracted from the DY016 and DY009 datasets (see Section C). For each good unit, we construct a template by averaging all of its spikes to get a mean waveform. As we are only using a segment of the recording for this analysis, some units with less spikes have noisier templates (see Supplementary Figure 2). As can be seen in the figure, we find 4 putative cell-type clusters which seem to roughly correspond to narrow-spiking inhibitory cells (green and blue) and broad-spiking, excitatory cells (yellow and red).

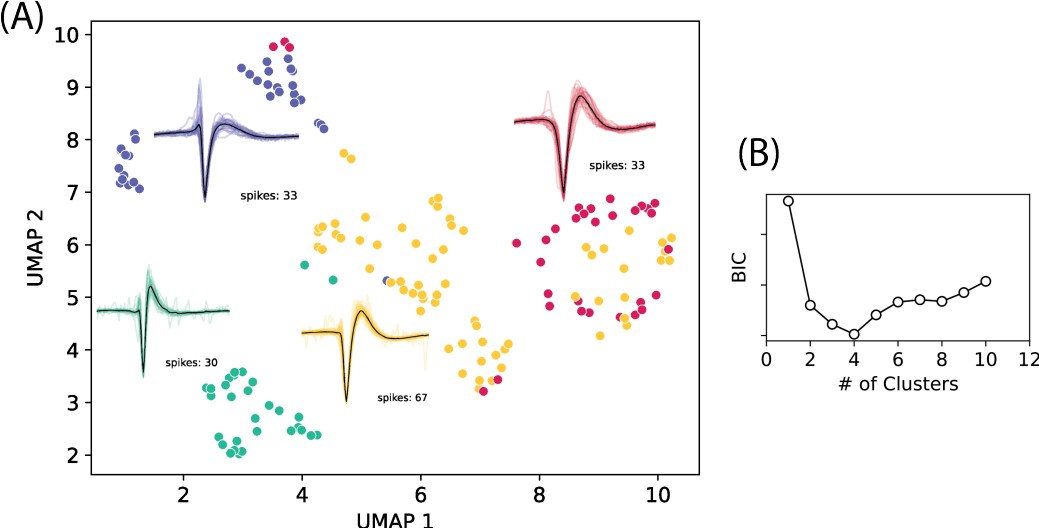

Supplementary Figure 2: **CEED cell-type classification on 163 IBL good units** We find 4 putative cell-type clusters which seem to roughly correspond to narrow-spiking inhibitory cells (green and blue) and broad-spiking, excitatory cells (yellow and red). This analysis is confounded by the template noise and the relatively small amount of neurons, however, CEED should have some robustness to noisy templates due to its data augmentations.

## B.2 WaveMap extended results

In Supplementary Figure 3, we plot the extended results for the 6 clusters that CEED finds in the WaveMap dataset. As can be seen, the (AP) width and the trough-to-peak features appear to be varied across clusters while the peak ratio is highly variable.

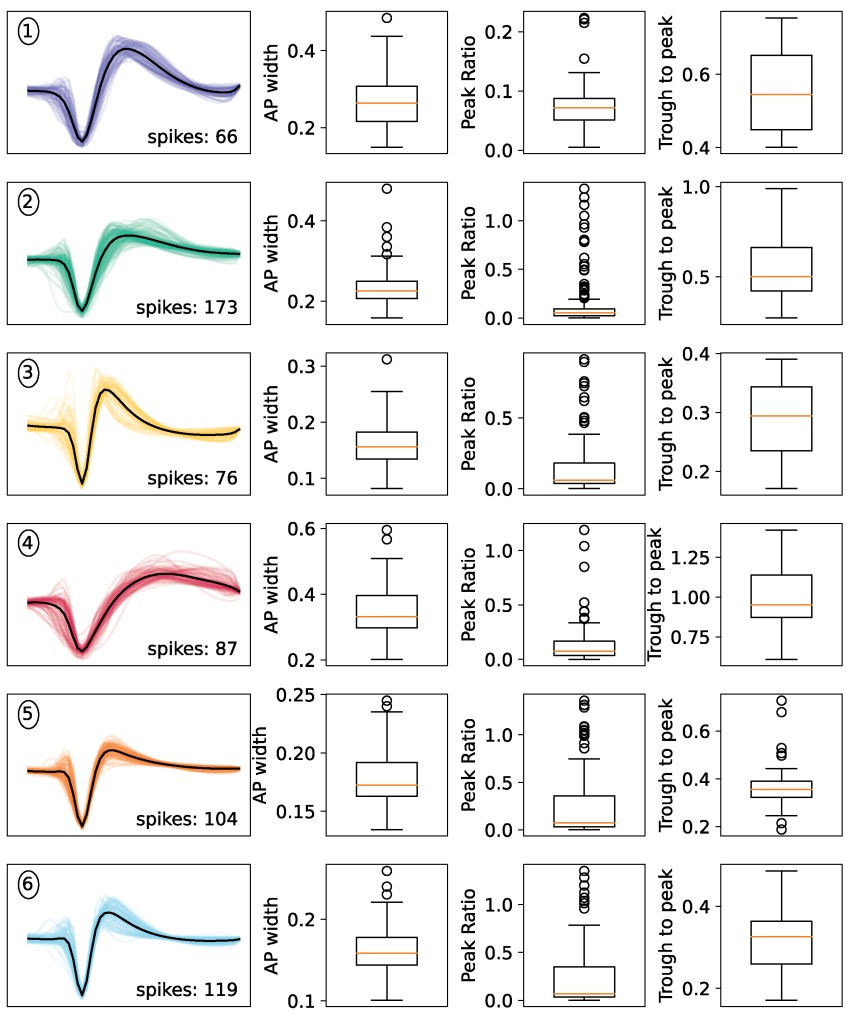

Supplementary Figure 3: **CEED cell-type classification WaveMap extended results**. In this figure, we visualize each of the 6 putative cell-type clusters found in the WaveMap dataset and plot some morphological features for each cluster. Both the action potential (AP) width and the trough-to-peak features appear to be varied across clusters while the peak ratio is highly variable.

## C  Data sources and datasets

**Data sources**  We utilize two real data recordings: the IBL DY016 recording and the IBL DY009 recording. We collect the real data (DY016 and DY009) from a Neuropixels 1.0 IBL recording (sampling frequency of 30 kHz) that has been spike sorted by Kilosort 2.5 [5]. For the DY016 recording, we extract spikes from a 36 minute chunk in which Kilosort 2.5 finds around 450 units. For the DY009 recording we take an 8 minute processed snippet where Kilosort 2.5 finds 464 units. In both recordings, we restrict our analysis of 10 neuron datasets to units deemed as 'good' which are units that pass IBL's quality metrics [1]. There are 81 good units in the DY016 dataset and 85 in the DY009 dataset. For the 400 neuron dataset we expand the neurons used to a pool of the good units and other qualitatively good looking units, i.e. having a higher spiking amplitude, smooth template, nice extracted spikes. For all datasets, we extract a spike train from these recordings and align the spiking events based on the trough of the detected spikes.

| Method | 400 neuron train set 10 neuron ID (ARI) | 390 neuron train set 10 neuron OOD (ARI) |
|---|---|---|
| CEED (200 spikes, 11 channels) | **.82 ± .07** | **.77 ± .08** |
| CEED (200 spikes, 5 channels) | .73 ± .09 | .74 ± .05 |
| Denoised PCA (200 spikes, 11 channels) | .43 ± .03 | .44 ± .04 |
| Denoised PCA (200 spikes, 5 channels) | .49 ± .04 | .49 ± .04 |

Supplementary Table 7: **Benchmarking CEED, and denoised PCA on in-distribution (ID) and out-of-distribution (OOD) data using a much smaller training set.** For evaluation, we fit 50 GMMs to the embeddings and compute the mean and std. of the adjusted rand index (ARI). First column: we train and test each method with spikes from 10 neurons. Second column: we train each method with spikes from 400 neurons and then test on the original 10 neurons which are **included** in the training set (ID). Third column: we train each method on spikes from 390 neurons and test on the original 10 neurons which are **not included** in the training set (OOD). This table shows that CEED is able to sustain its results even when the training set size drops significantly, where here the CEED models are trained 200 spikes from each unit as opposed to 1200 spikes from each unit in table 1 in the paper. All CEED results are generated using an MLP.

## D  Further CEED 400 neuron model benchmarking and performance

In this section, we evaluate the spike sorting generalization of a version of CEED that we train with 400 units and that is using the SCAM encoder. To evaluate the generalization performance, we extract subsets of units with size 3, 5, and 10 and perform a GMM clustering on the embeddings of CEED and of denoised PCA. As can be seen in Supplementary Figure 4, the average ARI of the clustering is much higher for CEED than denoised PCA.

To understand the CEED model further, we benchmark CEED models trained with an MLP encoder. Here we conduct three different forms of analysis: CEED models trained using much smaller training sets (200 spikes v. 1200 spikes), CEED compared to Denoised PCA when the dataset used has ground truth information, and the generalization of CEED to an unseen recording. For the first experiment, we construct a training dataset that uses spikes extracted from the same exact 400 neurons used to train CEED in the main text Table 1, but only include 200 spikes for each neuron in the training set rather than 1200 spikes per neuron. We can see from Supplementary Table 7 that CEED holds its performance on both 5 and 11-channel models for both In-distribution (ID) and Out-of-distribution (OOD) neurons when compared to CEED models trained on 1200 spikes, and still outperforms Denoised PCA by a significant margin. For the second experiment, we construct a dataset that makes use of ground-truth information that would not be available during spike-sorting and cell-type classification tasks. This ground truth information is the max amplitude channel of each neuron on the probe, and all waveforms extracted to create this dataset are centered on their putative neurons max channel. This type of dataset removes any shifts that might happen in the observed max amplitude channel for a particular waveform due to noise, etc., thus removing a significant nuisance variable in channel recordings. As we can see in Supplementary Table 8, a CEED model that does not use the max channel shift augmentation during training still outperforms Denoised PCA and a nonlinear baseline autoencoder on our GMM clustering metric. For the last experiment, we introduce an OOD recording, i.e. a recording from which we do not extract training or testing neurons. Instead, we use a CEED MLP model trained on the same 400 neuron, 200 spike dataset used for the first experiment above and test its ability to generalize to a completely unseen recording. From Supplementary Table 9, we can see that on all six test sets of good neurons that we extracted from this dataset CEED outperforms the Denoised PCA baseline on our clustering metric.

| Method (template max-channel aligned spikes) | 2D embedding | 3D embedding | 5D embedding | 10D embedding |
|---|---|---|---|---|
| CEED NoShift (11 channels, 200 spikes) | — | — | **.94 ± .04** | — |
| Denoised PCA (5 channels, 1200 spikes) | .49 ± .01 | .55 ± .07 | .70 ± .06 | .82 ± .08 |
| Denoised PCA (11 channels, 1200 spikes) | .43 ± .01 | .39 ± .02 | .44 ± .03 | .54 ± .03 |
| Autoencoder (5 channels, 1200 spikes) | — | — | .78 ± .04 | — |
| Autoencoder (11 channels, 1200 spikes) | — | — | .80 ± .09 | — |

Supplementary Table 8: **Benchmarking CEED, PCA, and an autoencoder on a template max-channel aligned version of the in-distribution (ID) data.** In this experiment, the data is constructed using *ground-truth* information. We use this information to center each spike on its template's max amplitude channel. This experiment is designed to evaluate the performance of each method without having to account for max channel shifts due to noise. For Denoised PCA and the autoencoders, we train with 1200 spikes from 10 neurons. For CEED, we train with 200 spikes from 400 neurons. We turn off the max channel shift augmentation for this dataset (i.e., Ceed NoShift). CEED and the AE were only trained for 5D due to time constraints. All CEED results are generated using an MLP architecture.

| 400 neuron, 200 spike train data | Test set (handpicked) | Test set 1 | Test set 2 | Test set 3 | Test set 4 | Test set 5 |
|---|---|---|---|---|---|---|
| CEED (11 channels) | .70 ± .05 | **.76 ± .05** | **.62 ± .03** | **.56 ± .02** | **.62 ± .04** | .61 ± .04 |
| CEED (5 channels) | **.72 ± .03** | .69 ± .03 | .51 ± .02 | .56 ± .02 | .52 ± .03 | **.66 ± .02** |
| DePCA (11 channels) | .53 ± .04 | .50 ± .02 | .40 ± .02 | .45 ± .03 | .43 ± .03 | .52 ± .03 |
| DePCA (5 channels) | .51 ± .06 | .58 ± .04 | .43 ± .04 | .51 ± .03 | .47 ± .03 | .54 ± .03 |

Supplementary Table 9: **Benchmarking CEED and denoised PCA on data from a completely unseen recording.** In this experiment, we collect and extract neurons from an IBL recording which the CEED model was not trained on and thus has not seen neurons from. We created six sets of 10 neurons for our evaluation with five of the sets chosen randomly (test sets 1-5) and one we handpicked for template diversity (test set (hand-picked)). We embedded the spikes for each test set using a CEED model trained on the 400 neuron, 200 spike dataset from the IBL recordings referenced in the paper. For Denoised PCA (DePCA) we extracted a training set from this unseen recording, which includes the neurons from all six test sets, denoised the waveforms, and fit PCA on the spikes from these neurons. We then embedded spikes from the six test sets into this PC space and test the clusterability of the embeddings by fitting 50 GMMs and computing the mean and standard deviation ARI. This experiment is designed to evaluate the performance of CEED on an OOD recording. We can see that CEED still outperforms the PCA baselines by a fair margin. All CEED results are generated using an MLP architecture.

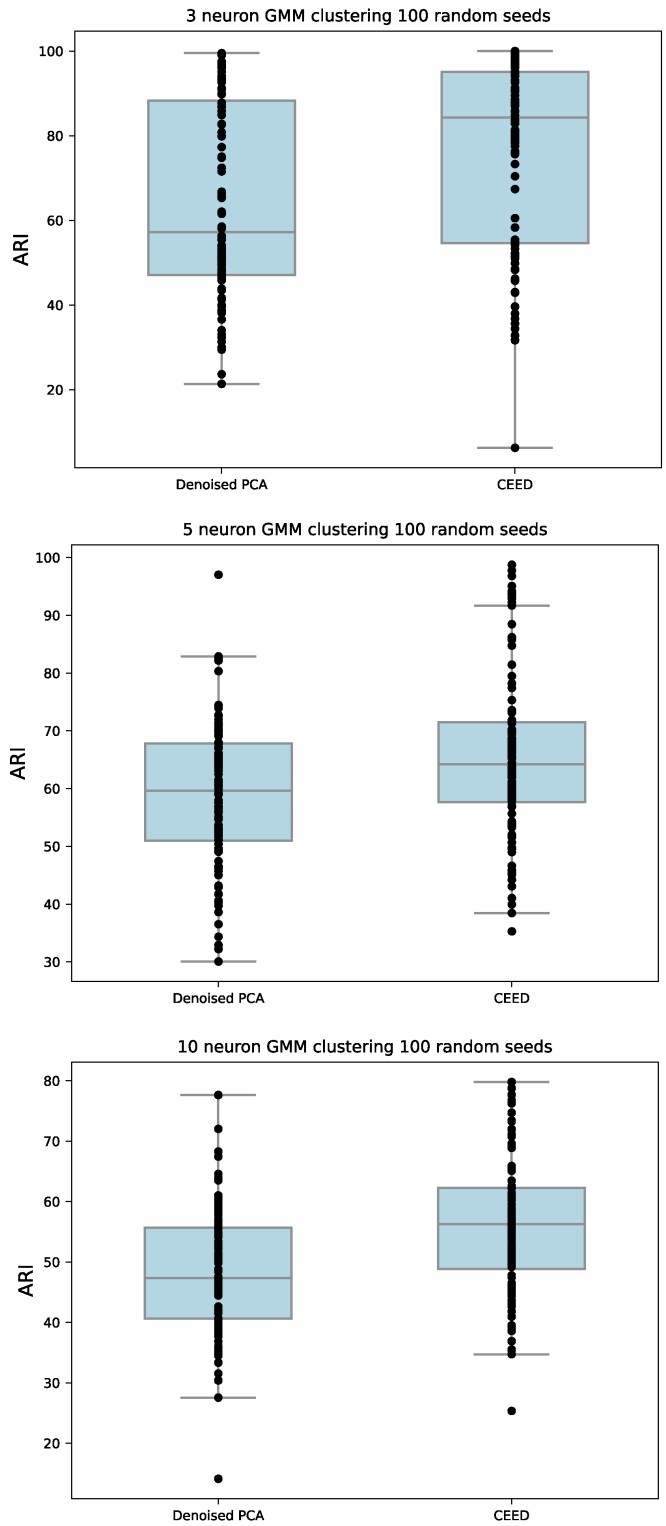

Supplementary Figure 4: **CEED SCAM 400 unit model evaluated on units subsets** In this figure, we show the results of a SCAM version of CEED trained on 400 units. To evaluate its spike sorting ability, we subsample sets of 3, 5, and 10 units with 100 random seeds and fit a GMM to each subset. We plot the ARIs for each clustering and compute the average ARI across all subsets. As can be seen the performance of CEED is much higher than denoised PCA all subset sizes.

# E    CEED ablation of the learned invariances

Analysis of ablations for each of the data augmentations used in CEED can be found in Supplementary Figure 5, where the different columns indicate varying levels of augmentations for the different augmentations, and the different rows show a set of three randomly selected neurons. We see that the representations do not change much when higher levels of augmentations are applied to spikes from all three neurons, thus graphically verifying that our representations are indeed invariant to the augmentations. Moreover, when we train CEED models without the augmentations referenced in Supplementary section A.1 we see that the performance of the model degrades quite a bit. In Supplementary Table 10 we can see that removing the noise, collision, and max channel shift augmentations all degrade performance of the model significantly. All results are generated using CEED trained with an MLP encoder.

|                                  | No max channel shift | No noise    | No collision | All augmentations |
| -------------------------------- | -------------------- | ----------- | ------------ | ----------------- |
| CEED (11 channels, 200 spikes)   | .44 ± .03            | .72 ± .06   | .60 ± .02    | **.82 ± .07**     |

Supplementary Table 10: **Data Augmentation Ablation**. For each column, we remove the associated data augmentation during training and benchmark the CEED model on the same 10 neurons as paper Table 1, Column 2. We train an MLP CEED with 200 spikes from 400 neurons for this analysis. Max channel shifting has the largest effect on CEED's overall performance. For an explanation of each data augmentation, see supplementary figure A.1. These CEED results are generated using an MLP architecture.

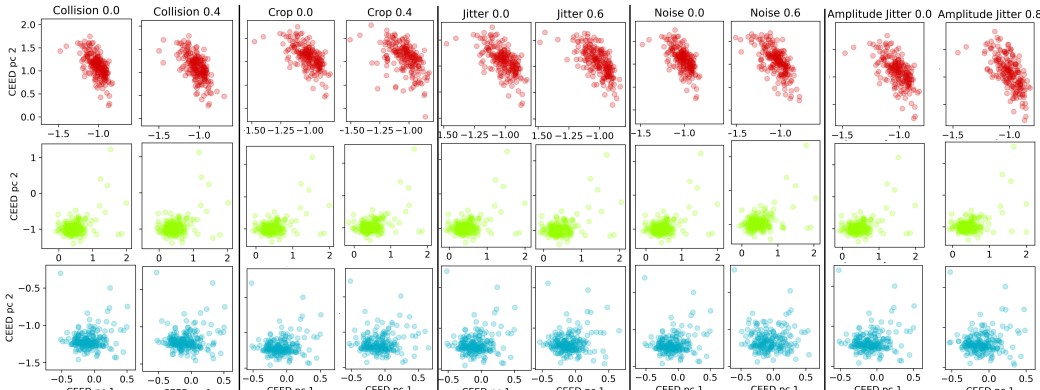

Supplementary Figure 5: **Visualizing CEED's appoximate invariances on 3 neurons**. In each row, we visualize a 2D PCA of the CEED embedding for three different neurons (red, green, and blue). Each set of two columns shows the CEED embedding before and after the data augmentation has been turned on. The numerical value in each column refers to the probability that the augmentation is applied to an individual spike. The embeddings are approximately invariant to the augmentations even when there is a high probability that the augmentation is applied. For an explanation of each data augmentation, see supplementary figure A.1.