# OpenReview forum: "Towards robust and generalizable representations of extracellular data using contrastive learning"
_NeurIPS.cc/2023/Conference — NeurIPS 2023 poster_

### Official Review · Reviewer_5ReK · 2023-06-27

**Soundness:** 4 excellent
**Presentation:** 3 good
**Contribution:** 3 good
**Rating:** 7
**Confidence:** 3

**Summary:**

The authors propose a contrastive learning method for obtaining representations of extracellular recordings which could be used for spike sorting and cell type classification. Transformer based encoder is used to generate low-dim representations of random views of spike waveforms which are then compared using a contrastive loss. The authors evaluate the proposed method on a simulated and real extracellular datasets, showing that it outperforms PCA embeddings for spike sorting and WaveMap on zero-shot cell types classification.



**Strengths:**

+ A nice application of modern machine learning methods (contrastive learning + transformers) to a classical problem (spike sorting)
+ Clear presentation

**Weaknesses:**

- I found the notation in Section 3 to be a bit confusing. E.g. \max over m and t in line 136 of A_{n, M} which doesn't have m and t in its indices. Superindices appear in line 149, even though W_{n, M} was defined without them in line 124. I think the notation could be simplified or at least made more consistent.
- The experimentation evaluation is limited. The authors show the proposed method outperforms PCA on spike sorting, but typically PCA is not the only tool used for spike sorting. I am not sure I understand how close CEED gets to state-of-the-art spike sorting methods or manual spike sorting based on a variety of tools (PCA, autocorrelations, visualisations, etc.)
- Ablation experiments could be useful as well. For example, it would be interesting to see if any of the proposed augmentations are more useful than the others or the effect of the transformer architecture choices.

**Questions:**

* How does CEED compare to state-of-the-art spike sorting methods? The ARI scores above 90 suggest it is quite similar to Kilosort in terms of spike sorting performance, would it be fair to say that CEED performs on par with state-of-the-art spike sorting methods?

* Have you tried comparing to non-linear embeddings rather than PCA (e.g. UMAP or autoencoders)? Would you believe CEED would outperform these methods as well?

**Limitations:**

I found the limitations section to be adequate and clearly written.

---

> ### Author Rebuttal · Authors · 2023-08-10
>
> We thank the reviewer for their strong assessment of our work and for their detailed comments and questions. The review is very useful and led us to some changes that strengthened the paper significantly. In the following response, we address each point raised by the reviewer.
> - The reviewer correctly pointed out that the notation in section 3 was sometimes confusing or inconsistent. We agree with this assessment and will make the notation more clear and consistent in the final version of the paper. Specifically, we will make sure the superscripts and subscripts are used consistently and correctly. We will also try to remove unneeded subscripts.
> - The reviewer pointed out that the experimentation evaluation is limited and that the baselines (PCA and denoised PCA) are insufficient given the rich literature of spike sorting methods. This is a valid concern and highlights a clarity issue with our paper. We want to emphasize that *CEED is not a spike sorting method, but rather a feature extraction method for spike sorting and cell-type classification.* While there are a number of spike sorting methods for MEAs, almost all of these methods utilize PCA for feature extraction (Klusta, HerdingSpikes2, Mountainsort4, SpykingCircus, Trideclous, and even Kilosort uses a form of SVD/PCA). We argue that any method that uses PCA would benefit from switching to our more discriminable and robust features. Since directly incorporating CEED into these methods is a non-trivial coding challenge (especially given the short rebuttal window), we plan to add a new experiment to the final manuscript where CEED is incorporated into a simple spike sorting pipeline which includes detection, featurization, clustering, and template matching. We will use SpikeInterface for this analysis.
> - The reviewer asked how CEED compares to a state-of-the-art spike sorting method such as Kilosort. It is important to note that Kilosort has a number of additional processing steps (cluster splitting and merging, template matching, etc.) which make direct comparison challenging. However, as stated in the above paragraph, CEED is a feature extraction method which can be utilized by full spike sorting pipelines.
> - The reviewer asked if we have compared CEED to any non-linear methods (i.e. umap or autoencoders). This is a great question as without any comparisons to non-linear methods it is unclear if the performance gains come from the non-linearity or the proposed data augmentations and contrastive learning objective. For the cell-type classification experiments, we refer the reviewer to Figure 3 where we provide a direct comparison to a non-linear method, WaveMap, which utilizes UMAP. We show that CEED is slightly stronger than WaveMap even without training on the dataset they provide. For spike sorting, we did not compare to a non-linear baseline in the submitted manuscript. To correct for this, we now include results in Table 1 of the attached PDF and the General Response showing that CEED also significantly outperforms a non-linear baseline (an autoencoder) on spike sorting. We thank the reviewer for this suggested experiment as it strengthens the manuscript considerably.
> - The reviewer mentions that ablation experiments would be useful. We completely agree and have included an ablation study of CEED’s data augmentations (Tables 2 and 3) in the attached PDF. We also benchmark an MLP architecture for CEED which has comparable performance to our SCAM architecture (see Table 1, Column 1).

---

> > ### Comment · Reviewer_5ReK · 2023-08-10
> >
> > Thank you very much for addressing my comments and providing additional results in the attached PDF. The rebuttal confirms my positive evaluation of the paper. I don't have any further questions at this point.

---

### Official Review · Reviewer_MJau · 2023-06-30

**Soundness:** 3 good
**Presentation:** 3 good
**Contribution:** 3 good
**Rating:** 6
**Confidence:** 4

**Summary:**

Rebuttal Update: I thank the authors for answering my questions and for the changes and additional experiments that they conducted. I have raised my score accordingly.

This paper proposes a contrastive framework to do spike sorting and cell type identification.

**Strengths:**

The paper is well written. It is easy to follow and the motivation is clear. The experiments are sound. The data augmentations encode very useful inductive biases for spike sorting.

**Weaknesses:**

The paper lacks more extensive benchmarking. There is a whole Zoo of spike sorting methods and dedicated benchmark (e.g., https://www.nature.com/articles/s41592-020-0902-0) – limiting the comparison to PCA and one recent method for clustering is not satisfactory.

Classical spike sorting builds on a lot of theory. Proposing a learned method instead is interesting, but it is important to check how this performs against classic methods more extensively out of distribution.

**Questions:**

It seems like the training data already requires knowledge of the spikes. Does this mean that the model cannot be trained unsupervised from MEA recordings alone?

Can you check if your learned features are actually becoming invariant to your data augmentations? I remember work showing that, in practice, this is often not the case.

Please give a more extensive discussion of refs 41-43. How do they differ from your approach? Could one just use CEBRA to do the same?

Is there no next token prediction in your transformer? If not, why do you use causal masks?

Line 95, ref 45 is form 2015 – please look at more recent reviews on Spike Sorting. I am skeptical that PCA is still ubiquitous, that seems like a straw man.

Lines 99-106: Please comment of functional, anatomical and genetic cell type classification approaches. Electrical profiles are only way.

116 -are. Also, this only gives approx. invariance (see above)

121 Please define MEA

145-147 please clarify

161 as -> a

177 is -> are

Are there no tokens in your transformer? I.e., no discrete symbols?

Why did you pick K=5? This seems low? Please comment on hyper parameter search.

190 SimCLR did not invent this loss function. Please cite the original references.

192 Did you perform an ablation on the projection MLP? What happens?

205 Only 10 neurons? Does your approach scale to big/relevant modern datasets (i.e., KILOsort...)?

227 If you have ground truth labels, why not compute accuracy (after optimal permutation with, e.g., Hungarian algorithm)?

**Limitations:**

I really want to know how classic spike sort algorithms (including the whole BSS pipeline and source recovery) compare to your approach as you are moving more out of distribution. My intuition is that learned (deep) approaches break down faster in those regimes.

---

> ### Author Rebuttal · Authors · 2023-08-10
>
> We thank the reviewer for their very detailed review and for their useful feedback. We hope to address their concerns point-by-point in the following response.
> - The reviewer pointed out that the manuscript lacks extensive benchmarking especially with the large number of spike sorting methods currently available for MEAs. This is a valid concern and highlights a clarity issue with our paper. We want to emphasize that *CEED is not a spike sorting method, but rather a feature extraction method for spike sorting and cell-type classification*. While there are a number of spike sorting methods for MEAs, almost all of these methods utilize PCA for feature extraction (Klusta, HerdingSpikes2, Mountainsort4, SpykingCircus, Trideclous, and even Kilosort uses a form of SVD/PCA). We argue that any method that uses PCA would benefit from switching to our more discriminable and robust features. Since directly incorporating CEED into these methods is a non-trivial coding challenge (especially given the short rebuttal window), we plan to add a new experiment to the final manuscript where CEED is incorporated into a simple spike sorting pipeline which includes detection, featurization, clustering, and template matching. We will use SpikeInterface for this analysis.
> - The reviewer correctly pointed out that the method needs more extensive benchmarking for out-of-distribution spike sorting datasets. We completely agree with this criticism and have added an experiment to quantify the performance of our method on out-of-distribution (OOD) data. We show that CEED outperforms all baselines (including a new non-linear autoencoder) on spikes from neurons outside the training set. Please see the Table 1 in the attached PDF and the General Response for a summary of this analysis.
> - The reviewer asked if CEED requires knowledge of the spikes and cannot be trained without supervision. CEED does require spike times and channel positions for all the spikes in the training set. These are easy to obtain experimentally. However, CEED does not require spike sorted data and can be trained after a simple voltage-based thresholding detection step. We agree it would be an interesting future work to find useful embeddings of extracellular data without spike detection.
> - The reviewer asked us to check if the proposed data augmentations actually induce approximate invariance in CEED’s representations. This is an important point and something we visualize in supplementary figure 6. We further extend this analysis in the attached PDF by adding a visualization of embeddings from 3 different neurons with all data augmentations (please see Figure 1).
> - The reviewer asked us to provide more discussion of refs 41-43 and how they differ from our method. References 41-43 all applied their method to neural response data (processed spike trains, obtained after spike sorting) and not to raw detections from extracellular data (before spike sorting). To our knowledge, we are the first contrastive learning method for extracellular data and had to develop specific data augmentations for this data modality. A large difference with CEBRA is that CEBRA explicitly mentions that they do not utilize data augmentations which could make the learned representations sensitive to the nuisance variables that are prevalent when working with MEAs (e.g. collisions). We would be happy to add this explanation to the final manuscript.
> - The reviewer asked about SCAM’s causal mask. As each next token in SCAM is conditioned on all the previous tokens and the last token is then used as the representation, we thought a causal mask would be appropriate.
> - The reviewer asked if there are no tokens in our transformer, i.e., no discrete symbols. For CEED, the signal at each time point of the data corresponds to a single "token" in our model. However, this value is continuous unlike the discrete inputs to a language transformer. Correspondingly, instead of a look-up table typically used in language transformers to map the discrete tokens into a higher dimensional embedding, we simply use a trainable linear layer to map the signal at each time point to a high dimensional embedding.
> - The reviewer asked us to add discussion about functional, anatomical, and genetic cell-type classification. We will add this discussion in the final manuscript.
> - The reviewer asked why we chose K=5 for the representation. We chose K=5 because the clustering performance and our heldout KNN metric saturates at this dimensionality. We would be happy to add this analysis to final version of the paper.
> - The reviewer asked if we performed an ablation of the projection MLP. We found that removing the MLP hurt the performance of our method. We would be happy to include this analysis in the final manuscript.
> - The reviewer asked if CEED can be scaled to big/relevant modern datasets which have potentially 100s of neurons. It is important to note that the clustering step in many spike sorting algorithms is performed on a local spatial neighborhood of channels which has anywhere from 1-10 neurons. Therefore, CEED can be used with this approach to cluster large extracellular datasets. We test this hypothesis in supplement figure 5 where we show that CEED can be trained and tested on a much larger dataset (spikes from 400 neurons) and still has large performance gains over the baseline feature extraction methods. In Table 1 of the attached PDF, we also show that when CEED is trained on spikes from 100s of neurons, it is able to generalize well to OOD data.
> - The reviewer asked why we did not evaluate our spike sorting experiments using the accuracy after optimal permutation with the Hungarian algorithm. We can add this metric to the final manuscript, however, we do not expect the conclusions to change as this metric is very correlated with the ARI.
> - The reviewer pointed out a few errors in the writing (missed citations, spelling, and abbreviations) that we will address in the final manuscript.

---

### Official Review · Reviewer_CNrX · 2023-07-07

**Soundness:** 3 good
**Presentation:** 4 excellent
**Contribution:** 3 good
**Rating:** 7
**Confidence:** 3

**Summary:**

This paper presents a novel method for self-supervised learning of useful representations for data from extracellular, multielectrode recordings in electrophysiology. The method uses a transformer architecture with causal spatiotemporal attention masks, and contrastive learning based on a set of desirable and relevant invariances. The paper describes the proposed neural architecture and training method, and tests them on spike sorting and cell-type classification (two standard problems for this type of data), on synthetic as well as real data from publicly available databases.


**Strengths:**

- the design choices for the model (including the details of the invariances used for CL) are well explained.

- the proposed method performs well, beating denoised PCA in spike sorting and WaveMap in cell-type classification.

- the paper is overall well written.

- the limitations of the method are clearly signposted.


**Weaknesses:**

- as reported by the authors, the method is currently slow to train and it is untested for spike sorting on more diverse data with waveforms that could have very different shapes than those in the training data (which would be the most useful case for general usage).


**Questions:**

Cold you give some more information about the practical computational and data requirements for the method? In particular, I could not find exact details on the hardware used for training (and the time required). It would also be great to have an additional analysis showing for instance how the results in figure 2 change as the amount of training data is changed.


**Limitations:**

The limitations of the method are discussed adequately. The paper mentions potential environmental issues related to the heavy computational requirements for model training, as well as possible mitigation strategies.

---

> ### Author Rebuttal · Authors · 2023-08-10
>
> We thank the reviewer for their strong assessment of our work and for their useful questions/feedback. We hope to address their concerns and questions in the following response.
> - The reviewer correctly pointed out that CEED is untested for spike sorting of more diverse data with waveforms that have different shapes than the training data. We agree with the reviewer that our experiments, which focused on CEED’s in-distribution (ID) performance, were insufficient for showing the general usability of the method. To address this, we include a new experiment to quantify the performance of our method on out-of-distribution (OOD) data. To strengthen our baselines, we also benchmark a non-linear baseline (an autoencoder) to show that the increase in performance comes from more than just the non-linearity in CEED. Please see Table 1 in the attached PDF and Global Response section for a summary of these results. We thank the reviewer for proposing this suggested experiment as we believe it strengthens the submission significantly.
> - The reviewer asked about the computational complexity and hardware used for training our method. For our spike sorting experiments, we utilized 16 NVIDIA V100s in parallel. Our runtime was 35 seconds per epoch for the 10 neuron, 200 spike, 11 channel model. For the 400 neuron, 200 spike, 11 channel model, the runtime was 3.1 minutes per epoch.
> - The reviewer mentioned that the computational complexity of our method is a weakness of the model. We note that this concern was shared across many of the reviewers so we provide a detailed response in the Global Response section. To briefly address the point here, we agree that our originally proposed model had significant computational complexity which limited its applicability to new datasets. In order to address this we, (1) sped up the data augmentation significantly by only computing the augmentations on a small neighborhood of channels, (2) proposed an alternative MLP architecture that also has strong performance and is much faster to train (only requires 1 V100 GPU), (3) quantified CEED on OOD datasets to show that even without training, CEED can outperform the current baselines for spike sorting or cell-type classification. While we are still looking into ways to improve the computational complexity of SCAM, this is challenging to address as transformers are still an active area of research. Recent progress in quantization [1] and acceleration software [2] offer promising solutions to transformers’ runtime issues, but incorporating these methods into CEED would be a future direction.
> - The reviewer mentioned that it would be useful to show how the results in figure 2 change as the amount of training data is changed. This is a great suggestion and we would be happy to include this analysis in the final manuscript. As can be seen in Tables 3 and 4 of our attached PDF, CEED still outperforms all baselines when using only 200 spikes from each neuron (rather than the 1200 spikes we use in figure 2).
>
> [1] Liu, Zhenhua, et al. "Post-training quantization for vision transformer." NeurIPS 2021
>
> [2] Ren, Jie, et al. "ZeRO-Offload: Democratizing Billion-Scale model training." 2021 USENIX Annual Technical Conference. 2021.

---

> > ### Comment · Reviewer_CNrX · 2023-08-15
> >
> > Thank you for answering my questions and for the additional work! I confirm my score.

---

### Official Review · Reviewer_xe16 · 2023-07-21

**Soundness:** 3 good
**Presentation:** 3 good
**Contribution:** 3 good
**Rating:** 5
**Confidence:** 2

**Summary:**

The paper proposes an approach based on transformer and contrastive learning for tackling problems from extra cellular recordings, including spike sorting and cell type classification. Experiments show the validity over standard PCA and traditional approaches in the field.

Note after rebuttal: I have appreciated the work on the revised manuscript, and raised the score accordingly. A concern on the rigor of the model selection is still present, though.

**Strengths:**

- The topic is fascinating
- The paper is well written
- The potential impact of the results are promising

**Weaknesses:**

- The experimental setup is not fully convincing

**Questions:**

- One of the main issues that I see in this work is also pointed out in the limitations section: the computational complexity of the approach. While the performance seems interesting, the point on the much higher computational complexity compared to the baselines risks to reduce the relevance of the analysis. Anyway, I suggest the authors to clearly indicate in the paper the number of trainable parameters, the computational complexity and an analysis of the computational cost required for training the proposed method, in comparison to the literature approaches.

- As far as I could see, no model selection on the hyper-parameters is performed.


**Limitations:**

I think the authors correctly indicated a major potential limitation of the approach, namely its huge computational costs compared to literature alternatives.

---

> ### Author Rebuttal · Authors · 2023-08-10
>
> We thank the reviewer for acknowledging that the paper is well-written, interesting, and that the results are promising. Since the reviewer has highlighted these strengths, we hope that by addressing their concerns, we can improve the rating.
> - The reviewer indicated that the experimental setup is not fully convincing. We agree with the reviewer that our spike sorting baselines and analyses could be improved. To address this, we include two new experiments to quantify (1) the performance of our method on out-of-distribution (OOD) data, (2) the performance of our method in comparison to a non-linear autoencoder. Please see Table 1 of the attached PDF and the Global Response section for a summary of these results. As morphoelectrical cell-type classification is still very much an open problem, we utilize the experimental setup and metric introduced in the Eric Kenji Lee 2021 paper. We feel this is a fair way of comparing our method to WaveMap. We do agree that quantifying our method and other cell-type classification methods with functional measures (cell-type selectivity, brain region cell-type distributions, etc.) is an exciting direction and would make for interesting future work.
> - The reviewer indicated that they did not see a section about model selection of the hyper-parameters. In supplement A.2, we discuss hyperparameters and how the learning rate was tuned using a heldout validation set (separate from the test set). We also did some hyperparameter tuning to choose the optimizer (‘adam’ vs. ‘sgd’) and representation size. We will include these details in the supplement of the final paper. We agree that more rigorous hyperparameter tuning could improve the performance of the model, however, we believe that it is a strength of the method that we already have strong results without too much hyperparameter tuning.
> - The reviewer expressed concerns about the computational complexity of our method. We note that these concerns were shared across other reviewers so we provide a detailed response in the Global Response section. To briefly address the point here, we agree that our originally proposed methods had significant computational complexity which limited their applicability to new datasets. In order to address this we: (1) sped up the data augmentation significantly by only computing the augmentations on a small neighborhood of channels, (2) proposed an alternative MLP architecture that also has strong performance and is much faster to train (only requires 1 GPU), (3) quantified CEED on OOD extracellular spikes to illustrate that even without training on new data, CEED’s features are more discriminable than the baselines’ features. While we are still looking into ways to improve the computational complexity of SCAM, this is challenging to address as transformers are still an active area of research. Recent progress in quantization [1] and acceleration software [2] offer promising solutions to transformers’ runtime issues, but incorporating these methods into CEED would be a future direction.
>
> [1] Liu, Zhenhua, et al. "Post-training quantization for vision transformer." NeurIPS 2021
>
> [2] Ren, Jie, et al. "ZeRO-Offload: Democratizing Billion-Scale model training." 2021 USENIX Annual Technical Conference. 2021.

---

> > ### Comment · Reviewer_xe16 · 2023-08-18
> > **Follow up after rebuttal**
> >
> > Thank you for your work. I have appreciated the work on the revised manuscript, and raised the score accordingly. A concern on the rigor (and completeness) of the model selection is still present, though.

---

> > > ### Author Response · Authors · 2023-08-19
> > > **Hyperparameter selection experiments**
> > >
> > > We greatly appreciate that the reviewer raised their score because of our additional experiments. Based on their response, however, we realize that not including hyperparameter/model selection experiments was a weak point of our original manuscript and rebuttal. To address this, we have run additional experiments to understand the effect of the *representation size*, *learning rate*, *batch size*, and *number of hidden layers in the MLP*. For all experiments, we compute the KNN decoding accuracy (common in contrastive learning literature) and our GMM clustering ARI metric on a heldout validation dataset of 10 neurons. Our training set consists of 200 spikes from 400 neurons for all the models. The results are summarized below.
> > >
> > > | Representation Dimension  | 2D | 3D  | 4D  | 5D (current) | 6D | 10D |
> > > |--------------------------|------------------------|-------------|-------------|------------|------------|------------|
> > > | KNN  | .63   | .88   | .94  | **.96**  | **.96**  | **.96** |
> > > | GMM  | .44 ± .01   | .67 ± .02   | .80 ± .06  | **.82 ± .07** | .79 ± .09  | .68 ± .04  |
> > >
> > > | Learning Rate | 1e-4 | 1e-3  (current) | 5e-3 |
> > > |--------------------------|------------------------|-------------|-------------|
> > > | KNN  | **.97**  | .96   | .95  |
> > > | GMM  | .81 ± .10  | **.82 ± .07** | .81 ± .06  |
> > >
> > > | Batch Size | 128 | 256 | 512 (current) |
> > > |--------------------------|------------------------|-------------|-------------|
> > > | KNN  | .95 | **.96**   | **.96**  |
> > > | GMM  | .80 ± .09  | .79 ± .08 | **.82 ± .07**  |
> > >
> > > | MLP Num Hidden Layers | 2 | 3 (current)  | 4 |
> > > |--------------------------|------------------------|-------------|-------------|
> > > | KNN  | **.96** | **.96**   | **.96**  |
> > > | GMM  | **.83 ± .09** | .82 ± .07 | .82 ± .09  |
> > >
> > > The MLP architecture we used for the rebuttal had a representation size of 5D, a learning rate of 1e-3, a batch size of 512, and 3 hidden layers. We hope these experiments bring confidence to the reviewer that our model choices were appropriate and that our model is not overly sensitive to any of hyperparameters.
> > >
> > > It is important to note that the GMM clustering performance does suffer as the dimensionality of the representation size becomes large (10D). We believe this is a limitation of our specific clustering algorithm, however, as the KNN decoding accuracy is still quite high at 10D.

---

### Author Rebuttal · Authors · 2023-08-10

We thank all the reviewers for the detailed and useful reviews. Using this feedback, we have made improvements to CEED and run a number of new experiments which we detail below. We also provide some discussion of shared reviewer concerns below. Please see the attached PDF for the referenced tables and figure.
- **Computational complexity**. There were concerns about CEED’s high computational complexity and the requirement that it needs multiple GPUs to train. To improve the computational complexity of CEED, we propose a few modifications to the original method. (1) We realized our original data augmentations were slower than expected because we were computing them on additional channels that were not used during training. By only computing the augmentations on a small neighborhood of channels, we see a noticeable per epoch runtime boost. (2) While our SCAM architecture showed high performance across a number of datasets, we want to emphasize that the CEED framework is general and can be utilized with other architectures that require less computational resources. To illustrate this, we train a number of MLPs using our data augmentation and contrastive learning scheme. We show that the performance of the MLPs are comparable to the original SCAM architecture (see Table 1, Column 1) with a significant reduction in computational complexity (see Table 2). We use the MLP architecture for all results in the attached PDF. While we are still looking into ways to improve the computational complexity of SCAM, this is challenging as transformers are still an active area of research. Recent progress in quantization [1] and acceleration software [2] offer promising solutions, but incorporating these methods into CEED would be a future direction.
- **Out-of-distribution (OOD) training and a new non-linear baseline**. A few reviewers wanted to see non-linear baselines and OOD performance evaluation for our spike sorting experiments. We completely agree with both of these points and have added a new experiment to address this (see Table 1).  In this experiment, we benchmark the performance of CEED, our original baselines, and a new non-linear autoencoder on a new 10 neuron OOD dataset. For this OOD dataset, we train each method with spikes from a large number of neurons (390) and then test on spikes from 10 heldout neurons (Table 1, Column 3). We show that CEED significantly outperforms the original baselines and the new autoencoder baseline for both the in-distribution (ID) and OOD datasets.
- **Ablations of the data augmentations**. Some of the reviewers wanted us to ablate our data augmentations to see which ones were the most important for CEED. Please see Table 3 of the attached PDF for an ablation of three of our data augmentations (all augmentations are detailed in supplement A.1). The most impactful data augmentation for CEED’s performance was the max channel shift augmentation (.38 drop in the ARI). To further test the importance of this max channel shift augmentation, we ran another experiment (see Table 4) where we created a new training and testing dataset. Instead of extracting each spike at its maximum amplitude channel; we instead extract each spike on *its template's maximum amplitude channel*. This means that spikes from the same neuron will be centered on the same channel. We designed this experiment to evaluate the performance of CEED and our baselines without having to account for max channel shifts due to noise. For fair comparison, we turn off  CEED’s max channel shift augmentation for this dataset. As can be seen, CEED still has the highest performance of all methods. We want to emphasize that this dataset utilizes ground-truth information (each neuron’s template) and is an ablation to better understand each method’s performance. We would be happy to add an ablation for all the data augmentations in the final version of the manuscript.
- **Visualizing approximate invariances of CEED's embeddings**. A few reviewers wanted to see if CEED's embeddings were approximately invariant to the proposed data augmentations. Please see Figure 1 in the attached PDF for a visualization of CEED embeddings for 3 neurons under all different data augmentations.
-  **CEED vs. full spike sorting pipelines**. A few reviewers had questions about how CEED compares to full spike sorting pipelines such as Kilosort. We want to emphasize that *CEED is not a spike sorting method, but rather a feature extraction method for spike sorting and cell-type classification*. While there are a number of spike sorting methods for MEAs, almost all of these methods currently utilize principal components analysis (PCA) for feature extraction (Klusta, HerdingSpikes2, Mountainsort4, SpykingCircus, Trideclous, and Kilosort use a form of SVD/PCA). We argue that any method that uses PCA would benefit from utilizing our more discriminable and robust features. Since directly incorporating CEED into these methods is a non-trivial coding challenge (especially given the short rebuttal window), we plan to add a new experiment to the final manuscript where CEED is incorporated into a simple spike sorting pipeline which includes detection, featurization, clustering, and template matching. We will use the SpikeInterface for this analysis.

[1] Liu, Zhenhua, et al. "Post-training quantization for vision transformer." NeurIPS 2021

[2] Ren, Jie, et al. "ZeRO-Offload: Democratizing Billion-Scale model training." 2021 USENIX Annual Technical Conference. 2021.

---

> ### Comment · Reviewer_MJau · 2023-08-14
> **IID vs OOD**
>
> Thank you for the OOD experiment! Can you please provide a bit more detail about what those 390 training and 10 test neurons are? Usually, splitting data into training and test (like it seems you have done here) would fall under IID (independent identically distributed). OOD (out of distribution), by contrast, refers to data that comes from a different distribution. For instance, training a classifier on dog vs cat images in summer and then testing it on winter images would be considered an OOD setting. I am interested in this because it would seem relevant to check if your method works also on experimental data that was, e.g., recorded in a different lab. Seeing it beat simple robust methods (like PCA) in those settings would be the most interesting test to assess your method's performance 'in the wild'.

---

> > ### Author Response · Authors · 2023-08-15
> > **IID vs OOD response**
> >
> > This is an excellent question and something we should have been more clear about in the rebuttal. In the original manuscript, we split our data by **spikes**. Since the training spikes and the test spikes came from the same neurons, we refer to this as in-distribution (ID). For our new out-of-distribution (OOD) experiment, we split our data by **neurons**. Since the training spikes and test spikes come from different neurons, we refer to this as OOD. Since each neuron is believed to have a unique spatiotemporal footprint on the probe (template), we thought that holding out neurons would address reviewer concerns about generalizability. All neurons in both the ID and OOD experiments are extracted from two International Brain Laboratory Neuropixels recordings with Kilosort 2.5.
> >
> > Another way to create OOD datasets is to split the data by **recording** (as you suggested in your response). This is a great idea and something we wanted to test. To test this, we have downloaded and extracted neurons from another IBL recording that was not used for training. We created six sets of 10 neurons for our evaluation with five of the sets chosen randomly (test sets 1-5) and one we handpicked for template diversity (test set (hand-picked)). We embedded the spikes for each test set using CEED, Denoised PCA (DePCA), and PCA. Each method was trained on 200 spikes from the 400 neurons we extracted from the two original IBL datasets. We again test the clusterability of the embeddings by fitting 50 GMMs and computing the mean and standard deviation ARI. The results are summarized in the table below.
> >
> > | 400 neurons train   | Test set (hand-picked) | Test set 1  | Test set 2  | Test set 3 | Test set 4 | Test set 5 |
> > |--------------------------|------------------------|-------------|-------------|------------|------------|------------|
> > | CEED (5 channels, 200 spikes)   | **.72 ± .03**              | .69 ± .03   | .51 ± .02   | **.56 ± .02**  | .52 ± .03  | **.66 ± .02**  |
> > | DePCA (5 channels, 200 spikes)  | .51 ± .06              | .58 ± .04   | .43 ± .04   | .51 ± .03  | .47 ± .03  | .54 ± .03  |
> > | PCA (5 channels, 200 spikes)    | .53 ± .04              | .31 ± .02   | .22 ± .02   | .34 ± .04  | .22 ± .02  | .26 ± .02  |
> > | **CEED (11 channels, 200 spikes)**  | .70 ± .05              | **.76 ± .05**   | **.62 ± .03**   | **.56 ± .02**  | **.62 ± .04**  | .61 ± .04  |
> > | DePCA (11 channels, 200 spikes) | .53 ± .04              | .50 ± .02   | .40 ± .02   | .45 ± .03  | .43 ± .03  | .52 ± .03  |
> > | PCA (11 channels, 200 spikes)   | .50 ± .03              | .30 ± .02   | .14 ± .01   | .30 ± .03  | .20 ± .01  | .28 ± .02  |
> >
> > As can be seen in the table, CEED still outperforms the PCA baselines by a fair margin on the OOD recording. We hope that this analysis is convincing and shows that when CEED is trained on a Neuropixels recordings, there is evidence that it can generalize to other Neuropixels recordings. At the moment, CEED would have to be re-trained to be used with a new probe geometry. Making CEED generalizable to new probe geometries would be an interesting future direction.
> >
> > We wanted to add one last comment about the OOD experiments. While we believe it is important to show the generalizability of CEED to new recordings, we also think it is important to note that both CEED and PCA can be trained/fine-tuned on new datasets. Especially now that the runtime of CEED is lower than before (please see Table 2 in the attached pdf), many users may choose to train CEED directly on their recordings and then use the spike embeddings for their downstream analysis (spike sorting, cell-type classification, etc.). This setting, where all neurons are included in both training and testing, would be ID.

---

> > > ### Comment · Reviewer_MJau · 2023-08-15
> > > **Thank you!**
> > >
> > > Excellent, thank you very much for the additional work. The results look good and built up confidence about the generalizability of the method. I will raise my score accordingly.

---

### Decision · Program_Chairs · 2023-09-21

**Decision:**

Accept (poster)

**Comment:**

The paper proposes a self-supervised feature extraction technique for spike sorting from extracellular electrophysiological recordings. The reviewers agree that it’s a novel and technically solid approach. They criticize the simple PCA baseline and somewhat limited evaluation. The authors have addressed some of these concerns in the rebuttal. In addition, spike sorting pipelines are usually quite complex and spike sorting is inherently difficult to benchmark quantitatively. The authors’ work concerns only the feature extraction step. Hence, I consider the present work a potentially valuable contribution that should be accepted.